# Skim Milk as a Multifunctional Cryoprotectant for Fish Probiotic *Enterococcus* spp.: Impact on Viability During Lyophilization and Long-Term Storage

**DOI:** 10.3390/microorganisms13112486

**Published:** 2025-10-30

**Authors:** Matheus Gomes da Cruz, Ana Maria Souza da Silva, Karen Dayana Prada-Mejia, Hector Henrique Ferreira Koolen, Guilherme Campos Tavares, Gustavo Moraes Ramos Valladão

**Affiliations:** 1Postgraduate Program in Biodiversity and Biotechnology, University of the State of Amazonas (UEA), Manaus 69065-001, Brazil; juliana-lester@hotmail.com (A.M.S.d.S.); hkoolen@uea.edu.br (H.H.F.K.); 2Laboratory of Parasitology and Pathology of Aquatic Organisms, Nilton Lins University (UNL), Manaus 69058-030, Brazil; karen.pradamejia@yahoo.com; 3Postgraduate Program in Aquaculture, Nilton Lins University (UNL), Manaus 69058-030, Brazil; gcamposvet@ufmg.br; 4Multi-User Center for the Analysis of Biomedical Phenomena, University of the State of Amazonas (UEA), Manaus 69065-001, Brazil; 5Department of Preventive Veterinary Medicine, School of Veterinary Medicine, Federal University of Minas Gerais (UFMG), Belo Horizonte 31270-901, Brazil

**Keywords:** aquaculture, cryoprotectant, lactic acid bacteria, probiotics, simulated gastrointestinal conditions, storage stability

## Abstract

This study investigated the efficacy of different cryoprotectants (maltodextrin, skim milk, trehalose, sucrose, fructose, and dextrose) in protecting probiotic cultures isolated from the microbiota of arapaima (*Arapaima gigas*) (*Enterococcus faecium* CRBP46 and *Enterococcus gallinarum* CRBP19) during lyophilization, storage (−25 °C, 4 °C, and 25 °C for 120 days), and exposure to simulated gastrointestinal fluids (SGF). Cell surface hydrophobicity and the ultrastructural aspects of the coating matrices were also evaluated. Skim milk, trehalose, and dextrose (only for *E. gallinarum*) protected *Enterococcus* spp. against the negative effects of lyophilization, resulting in minimal viability loss (≤0.03 log CFU/g) and ≥99.50% survival. All cryoprotectants promoted stability (≥8.87 log CFU/g) for both bacteria when stored at refrigeration and freezing temperatures for 120 days. However, only skim milk maintained high viability (≥6.83 log CFU/g) for *Enterococcus* spp. during 120 days of storage at room temperature. Additionally, *Enterococcus* spp. lyophilized with skim milk demonstrated stability in SGF, with high cell viability (≥8.97 log CFU/g) and survival over 97%. Skim milk also significantly increased the cell adhesion capacity of *Enterococcus* spp., making them more hydrophobic. Scanning electron microscopy showed that *Enterococcus* cells were incorporated into the skim milk matrix and that its lower porosity directly contributed to the preservation of cell viability. Therefore, we conclude that skim milk is the most effective cryoprotectant under the tested conditions for *E. faecium* and *E. gallinarum*, as it ensured stability and high viability for both bacteria throughout all post-lyophilization challenges, maintaining bacterial concentrations above those suggested for probiotic formulations. Our findings provide unprecedented insights into the development of long-term stable, dry autochthonous probiotics, aiming to strengthen a more sustainable aquaculture for the arapaima, the Amazon’s giant fish.

## 1. Introduction

Probiotics are live microorganisms that, when ingested in adequate doses and at appropriate times, can positively impact the host’s health [1]. However, to exert their beneficial effects on the host, it is essential that probiotics are consumed in daily concentrations of 7 to 9 log CFU/g or mL, and that minimum concentrations (6 log CFU/g or mL) of these microorganisms remain viable until they reach the intestine, the target organ for modulation in the host [2,3]. Lactic acid bacteria (LAB) are widely recognized as the most common probiotic microorganisms and are used in nutraceuticals, dairy products, fermented foods and agriculture [4]. Currently, there is a diversity of LAB that are suitable for probiotic formulations, and the genus *Enterococcus* stands out for having some species that are ideal for biotechnological applications in medicine (human and animal) and the food industry [5]. Nevertheless, despite technological advances, the development of probiotic products remains challenging, especially when aiming to obtain a stable formulation that preserves its functional properties and maintains high viability during long storage periods [6].

In the probiotic industry, various drying techniques are being developed and tested to promote the viability of probiotic microorganisms, aiming for both an effective oral delivery system and an extended shelf life for probiotic cultures [7]. For this purpose, lyophilization is employed as a gentle and effective drying technique to obtain dried probiotic cultures [8,9]. Additionally, lyophilization reduces deterioration caused by bacterial activity and protein denaturation [3,7,10]. Nevertheless, despite being a rapid and effective technique for dehydrating bacterial cells, lyophilization presents challenges, since the freezing and intense dehydration used to remove moisture from the cell suspension can form intra- and extracellular ice crystals [3,9,11]. This can lead to cell wall/membrane rupture, reducing the viability of the probiotic culture immediately after lyophilization or during prolonged storage [12]. However, these effects can be mitigated or prevented with the addition of cryoprotective molecules to the cell suspension before lyophilization, protecting cells from conditions adverse to their viability [13].

Cryoprotectants are molecules that reduce the negative impacts caused by lyophilization, such as cryoinjuries to the cell membrane, protein (including enzymes) denaturation and DNA damage [14,15,16,17,18]. Generally, cryoprotective agents act by increasing the osmotic pressure and viscosity of the solution [19], creating a glassy matrix with low molecular interaction that minimizes ice crystal formation [3,20,21]. Additionally, the efficacy of cryoprotective agents is directly associated with their ability to replace water molecules during the dehydration process [22], preserving the structure and functional integrity of cell membranes [23]. The appropriate selection of cryoprotective agents is essential for maintaining the viability of probiotic strains, as well as for protecting their biological characteristics, as different cryoprotectants possess distinct interactions with bacteria. Therefore, evaluating the interaction between the cryoprotectant and the bacterium is essential for selecting molecules that ensure greater cellular viability [9,12,13].

Cryoprotectants, such as maltodextrin, skim milk, trehalose, sucrose, fructose and dextrose, are widely used in probiotic formulations to obtain dried cultures, as they are considered safe, non-toxic, versatile and biodegradable molecules [7]. Previous studies have shown that probiotic strains lyophilized with these protective agents show greater viability and cell survival post-lyophilization [9,23], as well as during prolonged storage at different temperatures [11,24,25]. Moreover, an increase in the resistance of strains to simulated gastrointestinal conditions [7,25], as well as a greater capacity for adhesion to the intestinal mucosa [24], can also be observed.

The arapaima (*Arapaima gigas*) is a species of great economic and social importance for continental aquaculture in South America [26]. Currently, its production faces challenges from the excessive use of pharmaceuticals to control diseases [27,28], which can lead to the development of parasitic and bacterial resistance, environmental risks, and food safety concerns. To ensure sustainability in production, the use of probiotics emerges as a promising alternative, as it assists in the effective management of culture systems and is environmentally safe for the entire food chain [29]. Recently, our research group isolated *E. faecium* CRBP46 and *E. gallinarum* CRBP19 strains from the intestinal microbiota of healthy arapaima [30]. However, the use of bacteria from the *Enterococcus* genus is controversial because [31], despite acting as probiotic microorganisms [32,33,34,35], some strains are considered pathogenic and capable of acquiring and transferring virulence factors and antibiotic resistance genes [36,37]. However, we highlight that strains *E. faecium* CRBPP46 and *E. gallinarum* CRBP19 were subjected to in vitro (e.g., hemolytic activity, antagonistic activity against pathogenic bacteria, and antimicrobial susceptibility profile) and in vivo assays, with their pathogenicity evaluated in healthy arapaima (safety test). Therefore, both strains were considered safe for arapaima [30].

In the last 5 years, the adoption of strains isolated from the host itself (autochthonous) has been highly desirable, since these microorganisms already naturally inhabit the intestinal tract of fish and act in their organic defense system [38,39]. Pereira [40] demonstrated that the intestinal microbiota of healthy arapaima is dominated by the phyla Proteobacteria, Fusobacteria and Firmicutes (the phylum that contains *Enterococcus*). Recently, our research group showed that dietary supplementation with the autochthonous strains *Enterococcus* spp. significantly increases the resistance of arapaima against *Aeromonas jandaei* (unpublished data). Furthermore, other researchers have shown that the use of autochthonous probiotic strains, such as *Lactococcus lactis* subsp. *lactis* and *E. faecium*, yields promising results in enhancing the growth performance, immunity, and resistance to aeromonosis in arapaima [33,35,41]. However, in in vivo applications, the strains were incorporated directly into commercial feed by spraying the culture medium containing the probiotic bacteria, without any protective measures. This approach highlights a critical knowledge gap concerning the biotechnological processes required to enhance the efficiency of this promising input for aquaculture.

To the best of our knowledge, no studies have been published focusing on the development of a dry nutraceutical additive with *E. faecium* and *E. gallinarum* for arapaima. However, for the successful development of a dried nutraceutical formulation, a deeper understanding of the interaction between these strains and potential cryoprotectants is required. While previous studies have evaluated various protectants, the specific effects on key functional properties of *Enterococcus* spp., and the structural basis for this protection remain poorly explored. Therefore, we examined the protective effects of six cryoprotectants (maltodextrin, skim milk, trehalose, sucrose, fructose, and dextrose) on the viability and survival of *E. faecium* and *E. gallinarum*. Additionally, to better understand the functional consequences of this protection, we analyzed the cellular adhesion capacity and stability of *Enterococcus* spp. under different conditions (storage temperature and simulated gastrointestinal fluids). To investigate the ultrastructural aspects of the cell–matrix interaction, scanning electron microscopy was also employed.

## 2. Materials and Methods

### 2.1. Materials

Tryptone soy agar (TSA) and Tryptone soy broth (TSB) culture media were commercially acquired from Neogen Ltd. (São Paulo, Brazil). Maltodextrin, D-(+)-trehalose, sucrose, fructose (levulose), dextrose (glucose), pepsin (1:10,000), sodium chloride (NaCl), monobasic sodium phosphate, dibasic sodium phosphate, xylene, hydrochloric acid (HCl) and sodium hydroxide (NaOH) were obtained from Êxodo Científica Ltd. (São Paulo, Brazil). Skim milk powder was acquired from Sinergia Cientifica Ltd. (São Paulo, Brazil), and bovine bile from Sigma-Aldrich (St. Louis, MO, USA). All reagents were of analytical grade.

### 2.2. Probiotic Strains and Preparation of Bacterial Inoculum

The lactic acid bacteria *Enterococcus faecium* CRBP46 (GenBank accession number PV364135) https://www.ncbi.nlm.nih.gov/nuccore/PV364135.1 (accessed on 5 April 2025) and *Enterococcus gallinarum* CRBP19 (GenBank accession number PQ394772) https://www.ncbi.nlm.nih.gov/nuccore/2811597283 (accessed on 5 April 2025) were isolated from the intestinal microbiota of healthy arapaima (*Arapaima gigas*) by our research group. These *Enterococcus* strains were subjected to in vitro assays (e.g., hemolytic activity, antagonistic activity against pathogenic bacteria, and antimicrobial susceptibility profile) and in vivo tests, with their pathogenicity being assessed in healthy arapaima (safety test). Therefore, both strains were considered safe for arapaima [30]. Stock cultures of *E. faecium* and *E. gallinarum* were maintained in microtubes at −80 °C in TSB containing glycerol (30 mL/100 mL, Sigma Aldrich, St. Louis, MO, USA).

*Enterococcus* spp. strains were reactivated on TSA and incubated in a bacteriological oven (EL101/3, Eletrolab, São Paulo, Brazil) for 48 h at 35 °C. The purity of the bacterial colonies was determined by observing their morphological aspects and Gram staining [39]. After confirmation of purity, each bacterial strain was individually inoculated into TSB (3L) and cultured at 35 °C for 24 h (stationary phase). Bacterial cells were harvested by centrifugation (Universal 320/320R, Hettich, Tuttlingen, Germany) at 5423× *g* (6 min at 4 °C) and washed twice with sterile phosphate-buffered saline (PBS, pH 7.4) to remove any residual culture medium. After washing, the pellets of *Enterococcus* spp. cultures were resuspended in 560 mL of in PBS (pH 7.4) to obtain a standard cell suspension (SCS).

### 2.3. Cryoprotectant Preparation and Lyophilization Procedure

The SCS of each *Enterococcus* strains were vigorously homogenized and divided into 80 mL aliquots. Subsequently, the cryoprotectants maltodextrin (MD), skim milk (SKM), trehalose (TL), sucrose (SU), fructose (FT), and dextrose (DT) at 20% (*w*/*v*) were aseptically added to and homogenized in each suspension. The mixtures were maintained at room temperature (25 °C) for 30 min. This step is essential to allow cell permeation and reduce osmotic shock before freezing [42]. Afterwards, each solution was fractionated into 1 mL aliquots, aseptically transferred to cryotubes (5 mL), and frozen for 2 h (−80 °C), following the recommendations of Mafaldo [12] with minor modifications. A PBS was used as a control (CT). Lyophilization lasted 40 h and was performed at –50 °C ± 1 using a benchtop lyophilizer (LJJ, JJ Científica, São Paulo, Brazil). The vacuum pressure was maintained at <300 mTorr (~40 Pa) throughout the drying cycle. After lyophilization, the cryotubes were closed with their external screw caps, which contained a silicone sealing ring, and the dried *Enterococcus* spp. cultures were packaged in Low-Density Polyethylene (LDPE) bags, manually sealed, and stored protected from light (see Section 2.5). The cellular viability (log CFU/mL) of *Enterococcus* spp. before lyophilization was determined via serial dilutions of aliquots (100 µL) from each treatment in PBS (10^1^–10^6^) and plating on TSA using the spread plate technique [11]. Plates were incubated for 48 h at 35 °C, and colony-forming units (CFU) were enumerated by selecting plates containing 30–300 colonies [9]. The limit of detection (LOD) and limit of quantification (LOQ) of CFU/g were standardized for the entire study based on the initial dilution (10^−1^) [12]. The LOD (2 log CFU/g) was defined as the minimum detectable count (1 CFU) from a 100 µL plate inoculum, corresponding to 100 CFU/g. Additionally, the LOQ (3.5 log CFU/g) represents the minimum and reliable count of bacterial colonies (30 CFU) on the same plate, corresponding to 3.000 CFU/g.

### 2.4. Viability and Survival of Probiotic Strains After Lyophilization

After lyophilization, the dried *Enterococcus* spp. cultures with MD, SKM, TL, SR, FT, DT and CT were immediately rehydrated to their original volume (1 mL) with PBS (pH 7.4) and equilibrated at room temperature (25 °C) for 15 min [14]. Aliquots (100 µL) from each treatment were serially diluted in PBS (10^1^–10^6^) and plated on TSA using the spread plate technique [11]. The plates were incubated at 35 °C for 48 h under aerobic conditions, and colonies were enumerated [9]. The reduction in the number of viable bacterial cells after lyophilization was obtained using logN_0_-N_1_, where N_0_ represents the viable cell count before lyophilization and N_1_ corresponds to the number of viable cells immediately after lyophilization [43]. The survival rate (%) after lyophilization was determined using the formula: (A/B) × 100, where A and B represent the number of viable cells (log CFU/mL or g) after and before lyophilization, respectively [11].

### 2.5. Stability of Lyophilized Enterococcus *spp.* Powder Under Different Storage Conditions

The lyophilized *Enterococcus* spp. cultures from all treatments were evaluated under three different storage conditions: freezing temperatures (−25 ± 1.1 °C), refrigeration (4 ± 0.6 °C), and ambient temperature (25 ± 1.05 °C) for 120 days. The ambient relative humidity (36.50 ± 0.58%) was monitored throughout the storage period using a digital thermo-hygrometer (Incoterm^®^, Rio Grande do Sul, Brazil). Viable cell counts (log CFU/g) of both lyophilized strains were quantified every 30 days of storage, by serially diluting in PBS and plating on TSA to determine the number of viable bacterial cells after prolonged storage [9]. The reduction in the number of viable cells after prolonged storage at three different temperatures was obtained using log N_0_ − N_1_, where N_0_ represents the viable cell count immediately after lyophilization (0 days), and N_1_ corresponds to the number of viable cells for each monitored time interval (30, 60, and 120 days) and temperature (−25 ± 1.1 °C, 4 ± 0.6 °C, and 25 ± 1.05 °C) [43].

### 2.6. Simulated Gastrointestinal Fluid Exposure Test

The resistance of both lactic acid strains to different pH values (3, 5 and 7) and bile salt concentrations (0%, 0.4%, 0.7% and 1%) was evaluated after 12 months of storage at freezing temperature (−25 °C).

#### 2.6.1. Viability of Lyophilized *Enterococcus* spp. Under Simulated Gastric Conditions

The viable cells of *E. faecium* CRBP46 and *E. gallinarum* CRBP19 lyophilized with the cryoprotectants MD, SKM, TL, SR, FT, DT and CT were quantified after exposure to simulated gastric fluid, following the recommendations of Golnari [44], with minor modifications. Simulated gastric juice solutions were prepared using 0.5% (*w*/*v*) NaCl and 0.3% (*w*/*v*) pepsin. The pH of the NaCl solution was adjusted to 3, 5 and 7 using 1 N HCl or 1 N NaOH, and subsequently autoclaved. Pepsin was added to the different pH-adjusted solutions and sterilized using a membrane filter (0.22 µm). Lyophilized *Enterococcus* spp. powder (~0.1 g) was rehydrated in 1 mL of each simulated gastric juice (pH 3, 5, and 7), homogenized, and incubated for 4 h at 35 °C. Exposure of lyophilized *Enterococcus* spp. to the PBS solution (pH 7.4) was used as a quality control to ensure that the reduction in viable cell count (log reduction CFU/g) in each treatment resulted from exposure to acidic pH (3 and 5) and not from storage time. The number of viable cells (log CFU/g) was determined at 0, 120 and 240 min. The reduction in the viable cell count was obtained using log N_0_ − N_1_ [43].

#### 2.6.2. Viability of Lyophilized *Enterococcus* spp. Under Simulated Intestinal Conditions

The viability (log CFU/g) of *Enterococcus* spp. lyophilized with MD, SKM, TL, SR, FT, DT and CT was determined after exposure to bile salts [44], with modifications. The bile salt-containing solution was prepared using sterile PBS (pH 7.4) supplemented with different bile concentrations (0%, 0.4%, 0.7% and 1% *w*/*v*). The four solutions were sterilized using a membrane filter (0.22 µm). In summary, ~0.1 g of the lyophilized *Enterococcus* spp. powder was rehydrated in 1 mL of each bile salt solution (0%, 0.4%, 0.7% and 1% *w*/*v*), homogenized, and incubated for 5 h at 35 °C. Exposure of lyophilized *Enterococcus* spp. to PBS (pH 7.4, 0% bile) was used as a quality control to ensure that the reduction in viable cell count (log reduction CFU/g) in each treatment resulted from exposure to bile salts and not from storage time. The viability (log CFU/g) of both of the lyophilized strains in each treatment was determined at 0, 60, 180 and 300 min. The reduction in viable cell count was obtained using log N_0_ − N_1_ [43].

### 2.7. Microbial Adhesion to Hydrocarbons Test

The cell surface hydrophobicity (CSH) percentage of lyophilized *Enterococcus* spp. was determined based on the hydrocarbon adhesion method [16]. Probiotic powders from all treatments were individually rehydrated in PBS (pH 7.4) for 30 min and adjusted to an optical density (OD) of 0.5 ± 0.01 at 600 nm (OD_0_). Subsequently, 3 mL of the cell suspensions were supplemented with 1 mL of xylene (non-polar solvent), vigorously homogenized for 90 sec, and left to stand at room temperature (±25 °C) for 15 min. The hydrophobicity assay was performed with three independent replicates for each treatment.

After complete separation of the phases, the aqueous phase was carefully removed and evaluated (OD_1_). CSH was determined via the percentage reduction in optical density at 600 nm of the bacterial cell suspension resulting from the cell separation between the hydrocarbon and aqueous layers [16]. Cellular hydrophobicity (%) was calculated using the formula: [1 − (OD_1_/OD_0_)] × 100, where OD_0_ represents the initial OD_600 nm_ of the cell suspension for each treatment, and OD_1_ is the OD_600 nm_ of the aqueous phase after addition of xylene and subsequent phase separation.

Lyophilized *Enterococcus* spp. strains in all the treatments were classified as hydrophilic, moderately hydrophobic, or hydrophobic when bacterial adhesion was ≤20%, 21–50%, or ≥50%, respectively [45].

### 2.8. Scanning Electron Microscopy (SEM) Analysis of Lyophilized Enterococcus *spp.*

The morphological aspects and the microstructural properties of *E. faecium* and *E. gallinarum* were observed using a high-resolution scanning electron microscope (JSM IT500-HR Jeol, Tokyo, Japan) after 12 months of storage at freezing temperature (−25 °C). Lyophilized *Enterococcus* spp. samples from all the treatments were finely dispersed on an adhesive carbon tape (double-sided), mounted on stubs, and gold-coated using a sputter coater (DII-29010SCTR, Jeol, Tokyo, Japan). Images were recorded and captured with backscattered electron detection at an acceleration voltage of 10 kV and up to 3300× magnification.

### 2.9. Statistical Analyses

All the analyses were performed in triplicate, and data are presented as mean ± standard deviation. All the results were subjected to the Shapiro–Wilk and Levene’s tests for verification of normality and homogeneity, respectively. Data of post-lyophilization viability, viability under different storage conditions, and exposure to simulated gastrointestinal fluids were subjected to Fisher’s analysis of variance (ANOVA), and when significant differences were found between treatments, Tukey’s test (*p* < 0.05) was used for comparison of the means. Conversely, cell surface hydrophobicity data were analyzed using Welch’s ANOVA and, subsequently, subjected to Games–Howell’s test (*p* < 0.05) to compare differences between treatments. All the tests were performed with a significance level of 5% using Jamovi software, version 2.7.7 (Jamovi, Sydney, Australia).

## 3. Results

### 3.1. Impact of Cryoprotectants on the Viability and Survival of Enterococcus *spp.* After Lyophilization

*E. faecium* and *E. gallinarum* bacteria lyophilized with PBS solution (CT) and with cryoprotectants MD, SR and FT demonstrated a significant decline in viable bacterial cell counts post-lyophilization. However, we found that *E. faecium* lyophilized with SKM and TL, and *E. gallinarum* lyophilized with SKM, TL and DT, did not exhibit altered viability after lyophilization (*p* > 0.05).

The survival rate of *E. faecium* and *E. gallinarum* varied widely among all the treatments, with values ranging from 90.96% to 100.26%. However, the highest survival rates (≥99.50%) were observed only in the bacteria lyophilized with the cryoprotectants SKM, TL and DT (for *E. gallinarum* only), significantly surpassing other cryoprotectants and the control group (*p* < 0.05) (Table 1).

### 3.2. Influence of Storage Temperatures on the Viability of Lyophilized Enterococcus *spp.*

*E. faecium* and *E. gallinarum* bacteria lyophilized with the cryoprotectants SKM, TL, SR, FT and DT remained stable during long-term storage. Viable bacterial cell counts (≥8.17 log CFU/g) showed no significant changes (*p* > 0.05) in each treatment when comparing the samples of day 0 and day 120 under temperatures of −25 °C (±1.1) and 4 °C (±0.6) (Figure 1A,D; Figure 1B,E, respectively). However, both lyophilized bacteria, regardless of treatment, when stored at ambient temperature (25 °C ± 1.05) showed a significant decline (*p* < 0.05) in viability (overall reduction of up to 9.28 log CFU/g) when comparing the samples from day 0 and day 120 (Figure 1C,F).

Additionally, after 120 days, we observed that both bacteria stored at freezing temperatures (−25 °C ± 1.1) and refrigeration (4 °C ± 0.6) exhibited statistical differences (*p* < 0.05) between treatments. Nevertheless, despite these differences, we noted that *E. faecium* and *E. gallinarum* maintained viability equal to or greater than 8.17 log CFU/g for all treatments at these same temperatures, with minimal viability losses, ranging from 0.08 to 0.09 log CFU/g for −25 °C (±1.1) and 0.34 to 0.52 log CFU/g for 4 °C (±0.6), respectively (Figure 1A,D; Figure 1B,E). Conversely, when stored at ambient temperature for 120 days, *E. faecium* lyophilized with MD (−1.17 log CFU/g) and TL (−1.33 log CFU/g), and *E. gallinarum* lyophilized with MD (−9.22 log CFU/g), TL (−5.26 log CFU/g), SR (−5.79 log CFU/g) and DT (−9.25 log CFU/g), exhibited a significant decrease in viability (*p* < 0.05) in logarithmic units across all treatments (Figure 1C,F). However, we determined that, even under storage at ambient temperature, both lactic acid strains lyophilized with SKM remained stable and exhibited high viability, reaching 6.83 log CFU/g for *E. gallinarum* and 9.04 log CFU/g for *E. faecium* (Appendix A).

### 3.3. Influence of Cryoprotectants on the Stability of Lyophilized Enterococcus *spp.* Exposed to Different pH Values

The lactic acid bacterium *E. faecium* lyophilized with FT and CT showed a significant decline (*p* < 0.05) in viable bacterial cell counts, with values ranging from 0.80 log CFU/g and 0.92 log CFU/g, respectively. Conversely, we confirm that *E. faecium* lyophilized with the cryoprotectants MD, SKM, TL, SR and DT demonstrated stability and high viability (overall reduction of up to 0.22 log CFU/g) after 240 min of exposure to pH 3.

Additionally, although *E. gallinarum* exhibited significant differences between all the treatments (*p* < 0.05), the reduction in its viability did not exceed 0.11 log CFU/g after 240 min of exposure to pH 3 and pH 5 (Table 2).

The viability of both bacteria across all the cryoprotectant treatments was ≥8 log CFU/g, with a survival rate exceeding 90% after 240 min of exposure to pH 3 and pH 5 (Appendix A).

### 3.4. Impact of Cryoprotectants on the Viability of Lyophilized Enterococcus *spp.* Exposed to Bile Salts

When subjected to bile salt concentrations of 0.4%, 0.7% and 1%, *E. faecium* and *E. gallinarum* bacteria lyophilized with or without the cryoprotectants (MD, TL, SR, FT, DT and CT), showed a significant reduction (*p* < 0.05) in viable bacterial cell counts (log CFU/g) compared to the SKM treatment, after 300 min of exposure. Additionally, we observed that both strains lyophilized with SKM remained stable and exhibited minimal viability reduction (≤0.26 log CFU/g) after 300 min of exposure to different bile salt concentrations (Table 3).

The viability of *Enterococcus* spp. cultures lyophilized with SKM was ≥ 8.84 log CFU/g, with a survival rate of ≥ 98% after 300 min of exposure to 0.4%, 0.7%, and 1% bile salt concentrations (Appendix A).

### 3.5. Bacterial Adhesion to Hydrocarbons

The degree of cell surface hydrophobicity (CSH) of lyophilized *E. faecium* and *E. gallinarum* varied widely among all treatments, showing values between 29.86% and 87.08%; and 11.24% and 45.27%, respectively. However, we found that both LAB that were lyophilized with SKM exhibited the highest CSH values (45.27–87.08%) compared to the other cryoprotectants and the control group (*p* < 0.05) (Table 4). Additionally, based on the degree of adhesion to hydrocarbons, only lyophilization with SKM renders *E. gallinarum* and *E. faecium* moderately hydrophobic or hydrophobic, respectively. This marked increase in hydrophobicity suggests that lyophilization with SKM may enhance the strains’ potential to adhere to the gut mucosa, a key prerequisite for colonization.

### 3.6. Morphological Aspects of Lyophilized Enterococcus *spp.* Cultures

*E. faecium* and *E. gallinarum* bacteria lyophilized with PBS solution (CT) exhibited bacterial cells with a relatively smooth surface, a rounded shape, and slight morphological irregularities. The larger, cubic or angular particles are residues of crystallized PBS salts (Figure 2A and Figure 3A). Additionally, we observed the formation of bacterial biofilms and the presence of exopolysaccharide-like structures secreted into the extracellular matrix (Figure 2B and Figure 3B).

The treatments with MD and SKM showed fragments with irregular shapes and edges. Both treatments presented a surface with ridges and depressions, creating a rough and textured pattern (Figure 2C,D; Figure 3C,D, respectively). The TL treatment revealed a fragmented surface with irregular shapes and rounded edges. Additionally, the TL treatment resulted in a structure with microporosities and roughness (Figure 2E and Figure 3E). SR gave an irregular and porous matrix, with *E. faecium* and *E. gallinarum* bacterial cells being observed distributed at different points on the SR surface (Figure 2F and Figure 3F). The surface morphology of the samples lyophilized with DT is textured, with wavy and wrinkled patterns (Figure 2G and Figure 3G). The FT treatment showed spherical particles of varied sizes, widely distributed on the adhesive carbon tape (Figure 2H and Figure 3H). Moreover, we noted that the treatments with the cryoprotectants FT and DT were sensitive to the high-vacuum conditions used in SEM.

## 4. Discussion

Lyophilization is a relevant biotechnological preservation technique that enables the development of probiotic products in powder form. However, the intense dehydration of bacterial cells can cause structural and physiological damage to membranes, resulting in a loss of viability after application, which is exacerbated during storage [12,14,15]. Therefore, the addition of cryoprotective substances to the cell suspension before lyophilization is necessary in order to increase the viability of probiotic strains, minimizing the negative impacts resulting from lyophilization. This is the first study to investigate the cryoprotective potential of monosaccharide (FT and DT), disaccharide (TL and SR) and polysaccharide (MD) carbohydrates, and complex mixtures, such as SKM, in LAB isolated from the intestinal microbiota of arapaima, with the aim of developing an effective probiotic formulation. Our main results indicate that the cryoprotectants: (i) positively increased the viability of *E. faecium* and *E. gallinarum* after lyophilization; (ii) extended the shelf life under different storage conditions; (iii) improved the resistance of *E. faecium* and *E. gallinarum* to simulated gastrointestinal tract conditions; and (iv) enhanced the adhesion capacity of both bacteria to the intestinal mucosa.

Numerous studies have shown the effectiveness of various cryoprotective agents on the viability and survival of probiotic microorganisms after lyophilization [9,13,17,46]. Yuste [47] noted that *Lactiplantibacillus plantarum* subjected to lyophilization with skim milk, sucrose, trehalose and maltodextrin, either individually or in combination, maintained high viability (≥10.45 log CFU/mL). Similarly, *Lactobacillus plantarum* dried with skim milk, inulin, sucrose and maltodextrin are stable and exhibit high viability (≥8 log CFU/mL) and survival (31% to 91%) post-lyophilization [46]. Likewise, we can confirm that although there was a reduction in the viability of *E. faecium* and *E. gallinarum* lyophilized with MD, SR and FT, the lyophilization of both bacteria with the different cryoprotectants promoted adequate viability (≥8.60 log CFU/g) for potentially probiotic bacteria. Recently, Shu [8] proved that lyophilization causes significant alterations to cellular membranes (plasma membrane and cell wall), affecting their structure and functionality, as it forces bacterial cells to adapt to the water stress caused by the intense water removal during the drying process [48]. Moreover, considering that water interacts with macromolecules present in cellular membranes, its presence is essential for maintaining the structural and functional integrity of bacterial cells [3].

In this study, we suggest that the protection provided to *Enterococcus* spp. strains during lyophilization may be related to the natural adaptation of these microorganisms to stressful conditions but also results from the interaction exerted between these same strains and each cryoprotectant tested. In addition, the hydroxyl groups present in the molecular structure of MD, SKM, TL, SR, FT and DT replace sites previously occupied by water molecules before the dehydration process [49]. Consequently, this fact results in the stabilization of cellular membranes and the prevention of intracellular ice crystal formation [17], which, in turn, maintains all the biological structures in a similar form to those under hydrated conditions [50]. Thus, the selection of an ideal cryoprotectant is essential for the development of promising biotechnological products. Therefore, only the cryoprotectants SKM, TL (*E. faecium* and *E. gallinarum*) and DT (*E. gallinarum*) proved to be ideal for protecting *Enterococcus* spp. from the negative impacts of lyophilization, as they maintained a minimal reduction in viability (≤0.03 log CFU/g) and a high cellular survival rate (≥99.54%).

To our knowledge, studies investigating the impacts of temperatures that simulate real storage conditions for LAB, especially those of the genus *Enterococcus*, in probiotic formulations are limited. Generally, the storage of probiotic products at low temperatures is recommended, as it can maintain the survival of lyophilized probiotic cultures [45,51], since microorganisms are stored in a vitreous state [52], which results in the reduction in molecular activity and chemical reactions that could compromise the viability of bacterial cells [53].

In this study, cooling (4 °C) and freezing (−25 °C) temperatures offered the lowest losses of viability for lyophilized *E. faecium* and *E. gallinarum* cultures, which highlights the importance of maintaining both bacteria under these temperatures. Moreover, both temperatures caused minimal stress to our strains, ensuring high viability (≥8.17 log CFU/g) and stability for all treatments after 120 days. Recently, Mahmoodian [13] showed that *Pediococcus* sp. P15, *Weissella cibaria* ml6, and *Lactococcus lactis* ml3 lyophilized with a mix of cryoprotectants maintained constant viability rates (≥7 log CFU/mL), even after 365 days of storage at 4 °C. Similarly, *Lactobacillus rhamnosus* GDMCC1.325, lyophilized with cryoprotectants, maintained high stability for 60 days under refrigeration (≥8 log CFU/g) and freezing (≥9 log CFU/g) [51]. Therefore, long-term storage at freezing and cooling temperatures can significantly stabilize dried probiotic cultures. Under these conditions, the vitreous state, responsible for reducing intra- and extracellular molecular mobility [52], is maintained, preserving the natural structure of the cells and, consequently, resulting in high cellular viability after rehydration.

The search for protocols that ensure the stability of probiotic microorganisms at ambient temperature is highly desirable, as in addition to facilitating transport and storage [23], they are energetically economical and become more accessible in locations where temperature control is challenging or unavailable [3]. However, storing probiotic formulations at ambient temperature (≥25 °C) can be problematic, as it can increase humidity, water activity and chemical reactions, such as lipid oxidation, that are considered detrimental to cell survival [52,53]. Likewise, the generation of reactive oxygen species can contribute to damage to nucleic acids and cellular integrity, culminating in cell death [54].

This fact justifies the intense loss of viability (≤2 log CFU/g/detection limit) of *Enterococcus* spp. in treatments consisting of MD, TL, SR and CT for *E. gallinarum*; and FT and DT for both bacteria (*E. faecium* and *E. gallinarum*) over 120 days of storage at ambient temperature (~25 °C).

Additionally, we highlighted that the cryoprotectants SKM and TL presented the best protective performance immediately after lyophilization for *Enterococcus* spp. However, the protection provided by TL did not extend during storage at room temperature, which can be explained by their different protective mechanisms. Numerous studies have highlighted that the protection provided by TL is directly related to its ability to form a stable, non-crystalline glassy matrix that immobilizes cells and aids in water replacement [9,13,49,55,56]. However, this glassy state is sensitive to environmental conditions, especially temperature and humidity. Furthermore, when exposed to room temperature, TL can absorb moisture from the air, resulting in a reduction in its glass transition temperature and, consequently, its change from a glassy to a rubbery state [57,58]. This may result in a reduction in the protective capabilities of TL, as well as exposing *Enterococcus* spp. cells to moisture, oxidation, and chemical degradation, justifying the loss of viability. However, our lyophilized *Enterococcus* spp. strains with SKM showed high viability (6.83 and 9.04 log CFU/g for *E. gallinarum* and *E. faecium*, respectively) during the same period and temperature. Therefore, we believe that the combination of proteins (e.g., caseins and whey) and polysaccharides in SKM likely provided superior physical protection against moisture and temperature fluctuations, which ensured high viability of our strains during 120 days of storage at ambient temperature (~25 °C). This fact highlights the use of SKM as a versatile and the most effective cryoprotectant under the tested conditions, since ensuring the stability of live microorganisms at adequate concentrations throughout the entire shelf life of the product (probiotic powder) is essential for the preservation of its functional properties.

However, we emphasize that although our results highlight excellent stability for up to 120 days, especially for the probiotic formulation containing 20% skim milk, this period is considered insufficient to assess shelf life under commercial conditions, which requires prolonged evaluations of 12 months or longer. Therefore, future studies evaluating long-term storage are crucial to confirm these findings and validate the potential of our formulation for large-scale applications.

Maintaining high viability of probiotic microorganisms during gastrointestinal (GI) transit is essential for promoting their beneficial effects on the host [14,25,53,59]. Therefore, evaluating the number of viable cells of dried probiotic cultures exposed to acidic conditions and bile salts is crucial for confirming their ability to remain viable during GI transit [60], and that they reach the intestinal mucosa in sufficient quantities (6 to 8 log CFU/g) to adhere and colonize the host’s intestinal microbiota [3,53]. Furthermore, the survival of bacteria during GI transit results from the efficient preparation of the probiotic formulation (e.g., the concentration and type of cryoprotectant, as well as the drying protocol) [24], but it is also significantly influenced by storage conditions and storage period.

Recently, Guedes [14] exposed *Lactobacillus acidophilus* 05, *L. plantarum* 49 and *L. plantarum* 201 cultures lyophilized with β-glucan and fructooligosaccharides to simulated gastrointestinal fluids (pH 2, 1.2% bile salts) after lyophilization and 120 days of storage at 4 °C. The *Lactobacillus* spp. strains showed a significant reduction in viability, ranging from ~7.31 to 8.03 log CFU/mL (0 days) to 5.02 to 6.74 log CFU/mL (120 days). Similarly, *L. casei* ATCC 393, lyophilized with a hydrogel matrix (pea protein and alginate), showed greater tolerance to pH 2 (~8 log CFU/g) after lyophilization, when compared to *L. casei* stored for 84 days at −15 °C (~5 log CFU/g), despite maintaining viability during the storage period [61]. Additionally, both studies suggest that in vitro exposure tests to simulated gastric and intestinal conditions should be performed as soon as possible, as longer storage periods can make coating matrices more porous and facilitate H^+^ ion diffusion, making them more sensitive, regardless of storage temperature [14,61].

In our study, *E. faecium* and *E. gallinarum* maintained viability (≥8.45 log CFU/g) for 365 days at the freezing temperature (−25 °C) and were subsequently subjected to tests, demonstrating the stability of both lyophilized bacteria across all the treatments. Furthermore, even after storage and exposure to acidic conditions (pH 3) and bile salts (1%), our LAB lyophilized with all of the different cryoprotectants exhibited viable cell counts greater than the concentrations suggested for probiotic products. Nevertheless, only the probiotic formulations consisting of *Enterococcus* spp. and 20% SKM were stable and resilient (≥8.97 log CFU/g) in both stages, with a minimal reduction in viability (0.12 log CFU/g). This fact results from the excellent stabilizing and protective properties of SKM. Additionally, the high concentration of proteins present in SKM [62] provides a high buffering capacity in matrices containing SKM [59]. Thus, lyophilized probiotic cultures are more tolerant to environments with extreme pH conditions or bile salts, reducing or completely minimizing loss of cellular viability [25]. Therefore, based on our results, any stressful situations that reduce the viability of *Enterococcus* spp. cells during gastrointestinal transit were easily overcome in the presence of 20% SKM.

SKM is a high-molecular-weight cryoprotectant that acts by forming a robust and viscous layer around the bacterial cell [2]. This coating reduces the osmotic pressure between the intra- and extracellular environments [63], prevents the oxidation of proteins and enzymes [51] and damage to nucleic acids (DNA or RNA) [18], as well as reduces cell injuries [11,46] by stabilizing membrane components [51]. Additionally, the proteins present in skim milk (e.g., casein and β-lactoglobulin), stabilize the cell membrane through the interaction of hydroxyl and amine groups with the protein and lipid molecules present in the membrane [17,62], preserving its biochemical properties and metabolic activity [18,64]. Furthermore, during dehydration, the lactose molecules present in SKM form hydrogen bonds with the polar headgroups of membrane phospholipids, effectively replacing bound water [65]. Thus, this process prevents damaging phase transitions, impeding membrane leakage and the loss of cell function. Moreover, minerals present in SKM (e.g., calcium), may have contributed to stability by interacting with milk proteins, helping to encapsulate and protect the cells [66]. Therefore, we believe that the efficacy of SKM as a cryoprotectant for *Enterococcus* spp. resulted from the combined and interactive effects of its constituents.

Cell surface hydrophobicity (CSH) is one of the main characteristics to be investigated in microorganisms with probiotic potential, as it is associated with the strain’s ability to adhere to the host’s intestinal mucosa [16,45]. In addition, probiotic strains compete for adhesion sites on the surface of the intestinal epithelium [67]; therefore, strains with adhesive properties are able to colonize and persist in the intestinal microbiota, ensuring their beneficial effects on the host [6,45]. Petrova [68] demonstrated that *Enterococcus* spp. strains isolated from different environments exhibit CSH from 8.05 to 22.5%, based on the microbial adhesion to hydrocarbons method. Recently, our research group observed that the *E. faecium* CRBP46 strain showed no alteration in CSH before or after lyophilization [69], indicating that the freeze-drying process did not negatively impact the physico-chemical properties of the cell membrane of these bacteria.

The ability of probiotic strains to adhere to the intestinal mucosa is inherent to the adhesive properties of each strain [6,16,24,70]; however, some cryoprotectants can enhance CSH, increasing their adhesion capacity. Recently, Savedboworn [6] observed that *L. casei* TISTR1463 vacuum-dried with plant protein, fructooligosaccharides, galactooligosaccharides, inulin and sorbitol showed greater CSH (75.79% to 78.52%) when compared to the 0.85% NaCl (51.77%). Similarly, Archacka [24] noted that *Lactococcus lactis*, lyophilized or spray-dried with SKM, exhibited high affinity for Caco-2 and HT-29 cell lines, with adhesion of ~42 and 528 *L. lactis* cells per 100 epithelial cells, respectively. Likewise, we observed that SKM significantly increased the degree of CSH of *E. faecium* and *E. gallinarum* (45.27–87.08%) relative to the CT treatment (14.65–29.86%). Additionally, SKM acts as an extracellular cryoprotectant, forming hydrophobic bonds between proteins present in the skim milk itself, such as caseins [71] and β-lactoglobulin [70], and the surface bacterial cells, culminating in the stabilization of the probiotic–cryoprotectant interface [70]. Therefore, based on these findings, we suggest that the degree of CSH observed in our study was a result of the interaction of proteins present in SKM with both bacteria, making them more hydrophobic and, consequently, increasing their affinity for apolar surfaces.

Additionally, it is important to highlight that the assays conducted in our study, especially the simulated gastrointestinal fluid tolerance and cell surface hydrophobicity tests, were performed under in vitro conditions. Therefore, although our findings are promising indicators of the probiotic potential of *Enterococcus* spp. strains, as well as the protective efficacy of SKM, these cannot be extrapolated to predict colonization, persistence, and efficacy in vivo, especially in such a complex and dynamic environment as the intestinal microbiota of a live arapaima. Therefore, in vivo studies evaluating the effects of our *Enterococcus* spp. strains on growth performance, hemato-immunological responses, pathogen resistance, and their modulation in the native arapaima microbiota are essential to validate these findings and confirm the probiotic benefits in the target organism.

Scanning electron microscopy (SEM) is an analytical technique that allows for the observation of morphological details and ultrastructural aspects of the cryoprotectant-microorganism interface [72], as well as for verifying whether cryoprotectants effectively preserve bacterial structures during the lyophilization and storage process [15,52]. Generally, probiotic microorganisms lyophilized with cryoprotectants present bacterial cells distributed on the surface of the matrix, in interstitial spaces or within crypts formed during the drying process [6,12,14]; however, cells can also be incorporated by the cryoprotectants [11,15,73,74], which denotes biocompatibility. Recently, Li [11] lyophilized *L. curvatus* D2 with oleic acid, SKM, and TL, aiming to observe its ultrastructural aspects using SEM. However, *L. curvatus* was not detected in any coating matrix, suggesting that bacterial cells may not have been visualized due to the low magnification of the SEM or the incorporation of these microorganisms into the cryoprotectants. Similarly, *E. faecium* and *E. gallinarum* cells, lyophilized in matrices containing the cryoprotectants MD, SKM, TL, FT and DT, were also not identified at different magnifications. Nevertheless, through differential interference contrast (DIC) microscopy, the cells of both bacteria were clearly observed inside the matrices (Appendix A), indicating that our LAB were effectively incorporated into our different matrices. Therefore, the maintenance of viability observed in both bacteria may be related to the structural properties of our coating matrices, as they are compact and dense. This fact allows for a decrease in moisture retention and, consequently, in water activity within the matrices, which denotes protection and high viability across all the cryoprotectant treatments.

In this study, *E. faecium* and *E. gallinarum* lyophilized with PBS only (CT) exhibited biofilm formation, as well as the presence of exopolysaccharide-like structures (EPS) in the extracellular matrix. Recently, LAB *L. delbrueckii* subsp. *bulgaricus* CICC6098, *L. plantarum* 201, *L. acidophilus* 05 and *L. casei* 01 demonstrated intense EPS production after being subjected to adverse environmental conditions, such as dehydration and long-term storage [12,14,15]. Thus, we suggest that *Enterococcus* spp. produced and secreted EPS during deep freezing (−80 °C) and, subsequently, throughout the entire lyophilization process. EPS are carbohydrate molecules that are secreted extracellularly by various microorganisms [75]. Additionally, EPS are responsible for biofilm formation and maintenance, adhesion to surfaces (biotic and abiotic), and act as a temporary energy reserve for bacterial cells, being degraded when external nutrients are scarce [76]. Furthermore, EPS provide bacterial stability, protection, and resistance after exposure to different environmental stressors [77]. Therefore, based on these findings, we suggest that the maintenance of minimal viability of *Enterococcus* spp. in the CT treatment after lyophilization, prolonged storage at ambient temperature (for *E. faecium* only), and exposure to simulated gastrointestinal fluids resulted from endogenous factors of the probiotic bacteria themselves, but it was significantly enhanced by the presence of EPS.

## 5. Conclusions

This study presents, for the first time, the development of a dry and stable nutraceutical formulation from autochthonous probiotic bacteria from arapaima, aiming to create a promising and species-specific additive. We highlight that SKM acted as a multifunctional cryoprotectant for *E. faecium* and *E. gallinarum*. Our results demonstrate that the inclusion of 20% SKM in the cell suspension effectively protected the bacterial cells against the negative impacts of lyophilization, in addition to prolonging the shelf life of both bacteria at three different storage temperatures for up to 120 days, while maintaining bacterial concentrations above those suggested for probiotic formulations. The ability of SKM to maintain probiotics’ viability at room temperature for 120 days demonstrates a practical advantage for handling, transport, and short-term storage without the need for refrigeration. Furthermore, SKM proved effective in protecting *Enterococcus* spp. during adverse exposures to simulated gastrointestinal fluids, and it increased the adhesion capacity of these LAB to the intestinal mucosa. SEM revealed that *Enterococcus* spp. cells were incorporated into the SKM matrix, whose lower porosity directly contributed to the preservation of cell viability. Our findings highlight that the use of multifunctional cryoprotectants, such as SKM, is a low-cost and effective strategy to enable the use of host-specific probiotics in dry formulations, in addition to strengthening the development of a more environmentally sustainable aquaculture.

## Figures and Tables

**Figure 1 microorganisms-13-02486-f001:**
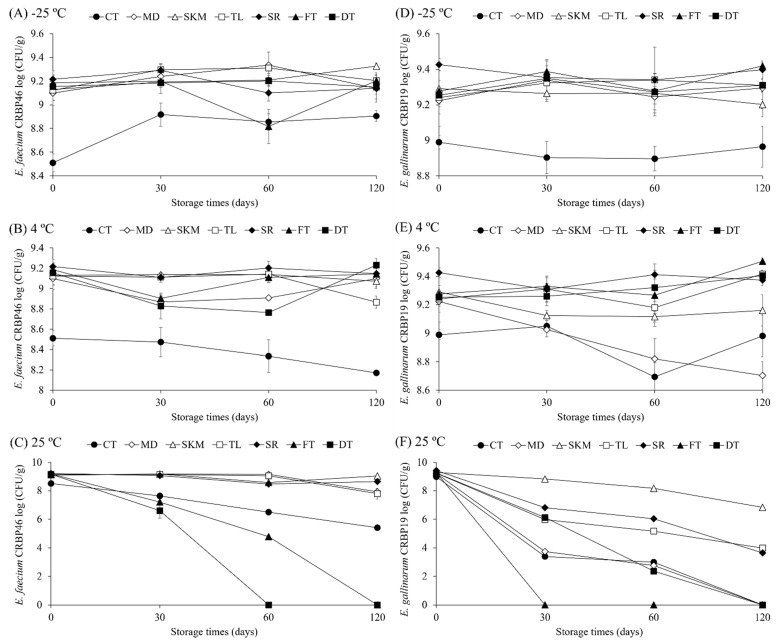
Influence of storage temperature (4 °C, 25 °C, and −25 °C) on cell viability (log CFU/g) of *E. faecium* CRBP46 (**A**–**C**) and *E. gallinarum* CRBP19 (**D**–**F**), freeze-dried with different cryoprotective agents over 120 days. CT: control (phosphate-buffered saline—PBS); MD: maltodextrin; SKM: skim milk; TL: trehalose; SR: sucrose; FT: fructose; DT: dextrose.

**Figure 2 microorganisms-13-02486-f002:**
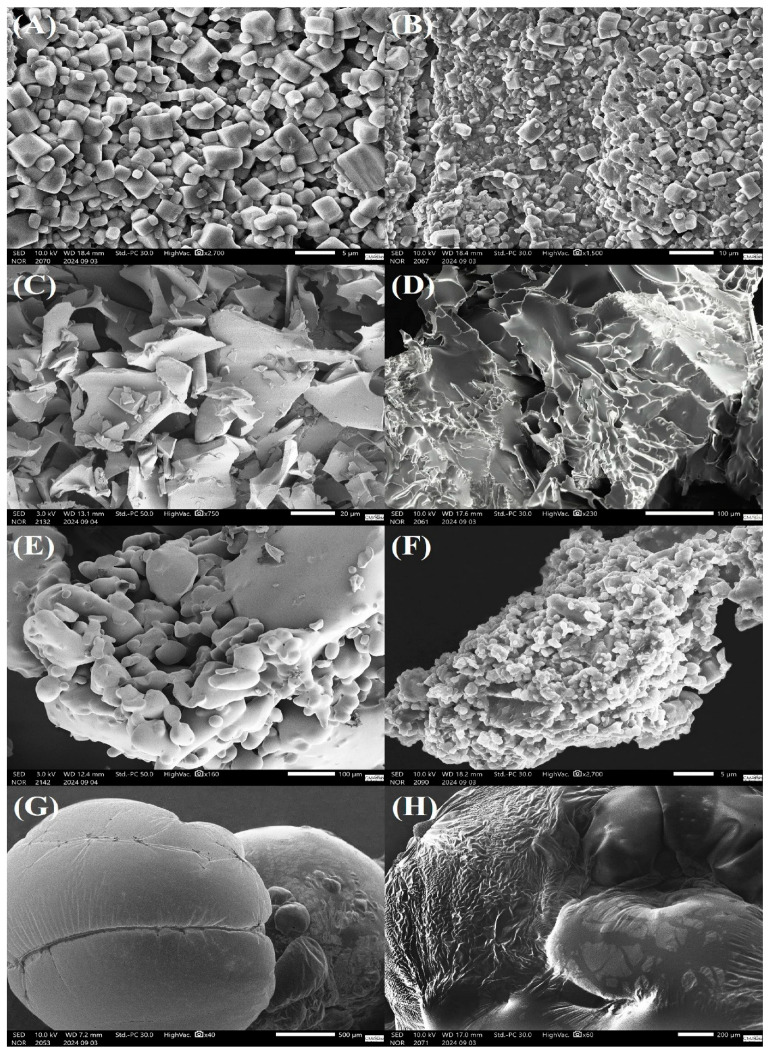
Micrographs of *E. faecium* CRBP46 cultures freeze-dried with different cryoprotectants: (**A**,**B**) CT, (**C**) MD, (**D**) SKM, (**E**) TL, (**F**) SR, (**G**) FT, (**H**) DT. CT: control (phosphate-buffered saline—PBS); MD: maltodextrin; SKM: skim milk; TL: trehalose; SR: sucrose; FT: fructose; DT: dextrose.

**Figure 3 microorganisms-13-02486-f003:**
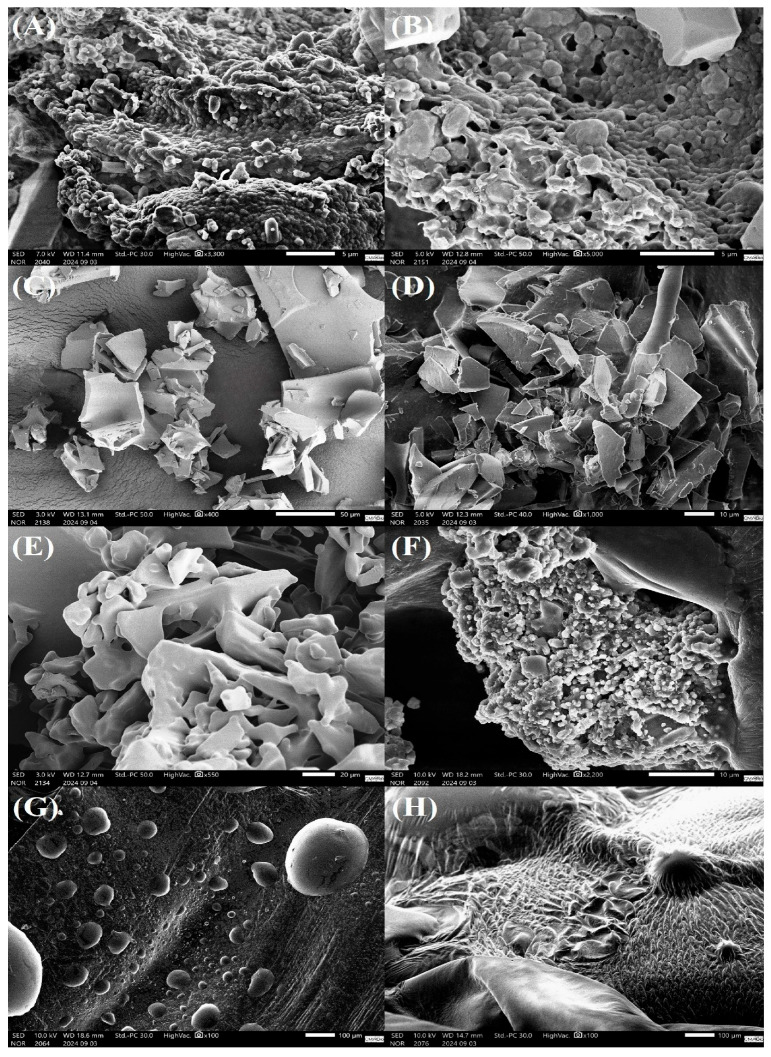
Micrographs of *E. gallinarum* CRBP19 cultures freeze-dried with different cryoprotectants: (**A**,**B**) CT, (**C**) MD, (**D**) SKM, (**E**) TL, (**F**) SR, (**G**) FT, (**H**) DT. CT: control (phosphate-buffered saline—PBS); MD: maltodextrin; SKM: skim milk; TL: trehalose; SR: sucrose; FT: fructose; DT: dextrose.

**Table 1 microorganisms-13-02486-t001:** Viability (Log CFU/mL or g) and survival rate (%) of *Enterococcus faecium* (CRBP46) and *Enterococcus gallinarum* (CRBP19) immediately after freeze-drying with different protective agents.

Probiotic Strains	Cryoprotector	Before Freeze- Drying (Log CFU/mL)	After Freeze- Drying (Log UFC/g)	Log Reduction (Log CFU/mL Before—Log CFU/g After Freeze-Drying)	Survival Rate (%)
*Enterococcus faecium*	Control	8.98 ± 0.01	8.17 ± 0.01	0.81 ± 0.02 ^A^	90.96 ± 0.18 ^C^
	Maltodextrin	8.93 ± 0.13	8.67 ± 0.07	0.27 ± 0.06 ^B^	97.03 ± 0.68 ^B^
	Skim milk	8.72 ± 0.03	8.72 ± 0.00	0.00 ± 0.02 ^C^	99.86 ± 0.19 ^A^
	Trehalose	8.77 ± 0.00	8.75 ± 0.01	0.02 ± 0.00 ^C^	99.78 ± 0.05 ^A^
	Sucrose	9.11 ± 0.03	8.68 ± 0.15	0.43 ± 0.12 ^B^	95.29 ± 1.31 ^B^
	Fructose	9.05 ± 0.04	8.61 ± 0.07	0.44 ± 0.11 ^B^	95.17 ± 1.16 ^B^
	Dextrose	9.00 ± 0.07	8.60 ± 0.01	0.40 ± 0.06 ^B^	95.56 ± 0.68 ^B^
*Enterococcus gallinarum*	Control	9.68 ± 0.08	9.11 ± 0.01	0.57 ± 0.07 ^A^	94.09 ± 0.68 ^C^
	Maltodextrin	9.66 ± 0.01	9.46 ± 0.00	0.20 ± 0.02 ^B^	97.96 ± 0.17 ^B^
	Skim milk	9.65 ± 0.00	9.64 ± 0.05	0.02 ± 0.06 ^C^	99.54 ± 0.43 ^A^
	Trehalose	9.51 ± 0.03	9.54 ± 0.02	−0.02 ± 0.00 ^C^	100.26 ± 0.04 ^A^
	Sucrose	9.66 ± 0.00	9.45 ± 0.01	0.21 ± 0.02 ^B^	97.83 ± 0.16 ^B^
	Fructose	9.68 ± 0.00	9.43 ± 0.03	0.25 ± 0.03 ^B^	97.41 ± 0.36 ^B^
	Dextrose	9.57 ± 0.06	9.53 ± 0.08	0.03 ± 0.02 ^C^	99.67 ± 0.22 ^A^

^A–C^ Different superscript capital letters in the same column denotes difference (*p* < 0.05) among the count’s loss of cells during freeze-drying based on Tukey’s test. Values represent mean ± standard deviation obtained from independent samples (*n* = 3). Control: Phosphate-buffered saline solution (PBS). Data of viable counts before and after freeze-drying are presented in wet basis.

**Table 2 microorganisms-13-02486-t002:** Logarithmic reduction (log CFU/g) in the viable cell count (mean ± standard deviation) of freeze-dried *Enterococcus* spp. exposed to acidic pH (5 and 3) for 4 h.

***Enterococcus faecium* CRBP46**			
Cryoprotector	Log reduction (log CFU/g pH 7–log CFU/g pH 5)	Log reduction (log CFU/g pH 7–log CFU/g pH 3)	Log reduction (log CFU/g pH 5–log CFU/g pH 3)
CT	−0.32 ± 0.19 ^D^	0.92 ± 0.06 ^A^	1.24 ± 0.13 ^A^
MD	−0.04 ± 0.06 ^BC^	0.12 ± 0.03 ^B^	0.16 ± 0.04 ^E^
SKM	−0.12 ± 0.03 ^BCD^	−0.12 ± 0.08 ^C^	0.00 ± 0.05 ^E^
TL	−0.22 ± 0.08 ^CD^	−0.02 ± 0.02 ^C^	0.20 ± 0.10 ^CDE^
SR	0.12 ± 0.04 ^AB^	0.22 ± 0.03 ^B^	0.10 ± 0.07 ^DE^
FT	0.03 ± 0.01 ^AB^	0.80 ± 0.02 ^A^	0.78 ± 0.01 ^B^
DT	−0.28 ± 0.08 ^CD^	0.12 ± 0.04 ^B^	0.39 ± 0.05 ^C^
***Enterococcus gallinarum* CRBP19**			
Cryoprotector	Log reduction (log CFU/g pH 7–log CFU/g pH 5)	Log reduction (log CFU/g pH 7–log CFU/g pH 3)	Log reduction (log CFU/g pH 5–log CFU/g pH 3)
CT	0.08 ± 0.16 ^A^	0.11 ± 0.09 ^A^	0.04 ± 0.10 ^A^
MD	−0.05 ± 0.09 ^AB^	−0.02 ± 0.12 ^AB^	0.03 ± 0.21 ^A^
SKM	−0.21 ± 0.07 ^AB^	−0.44 ± 0.11 ^C^	−0.23 ± 0.11 ^A^
TL	−0.07 ± 0.11 ^AB^	−0.11 ± 0.04 ^AB^	−0.04 ± 0.14 ^A^
SR	−0.33 ± 0.26 ^B^	−0.29 ± 0.10 ^BC^	0.04 ± 0.24 ^A^
FT	−0.02 ± 0.11 ^AB^	−0.20 ± 0.16 ^BC^	−0.17 ± 0.12 ^A^
DT	0.02 ± 0.01 ^AB^	−0.04 ± 0.05 ^AB^	−0.06 ± 0.06 ^A^

^A–E^ Superscript capital letters in the same column denote statistically different (*p* < 0.05) treatments of freeze-drying based on Tukey’s test. The values represent the mean ± standard deviation obtained from independent samples (*n* = 3). CT: control (phosphate-buffered saline—PBS); MD: maltodextrin; SKM: skim milk; TL: trehalose; SR: sucrose; FT: fructose; DT: dextrose. Negative values are interpreted as indicating the absence of loss of cell viability, suggesting tolerance after exposure to the stressful condition.

**Table 3 microorganisms-13-02486-t003:** Logarithmic reduction (log CFU/g) in the viable cell count (mean ± standard deviation) of *Enterococcus* spp. exposed to bile salts (0%, 0.4%, 0.7%, and 1%) for 5 h.

***Enterococcus faecium* CRBP46**						
Cryoprotector	Log reduction (log CFU/g 0%–log CFU/g 0.4%)	Log reduction (log CFU/g 0%–log CFU/g 0.7%)	Log reduction (log CFU/g pH 0%–log CFU/g 1%)	Log reduction (log CFU/g 0.4%–log CFU/g 0.7%)	Log reduction (log CFU/g 0.4%–log CFU/g 1%)	Log reduction (log CFU/g pH 0.7%–log CFU/g 1%)
CT	1.52 ± 0.05 ^B^	1.52 ± 0.06 ^AB^	1.74 ± 0.05 ^A^	0.00 ± 0.04 ^BC^	0.22 ± 0.02 ^AB^	0.22 ± 0.05 ^A^
MD	1.20 ± 0.08 ^C^	1.69 ± 0.17 ^A^	1.15 ± 0.12 ^B^	0.49 ± 0.12 ^A^	−0.05 ± 0.05 ^B^	−0.53 ± 0.09 ^B^
SKM	0.13 ± 0.10 ^E^	−0.03 ± 0.02 ^D^	0.03 ± 0.04 ^C^	−0.15 ± 0.08 ^C^	−0.10 ± 0.11 ^BC^	0.05 ± 0.04 ^A^
TL	1.00 ± 0.11 ^CD^	1.25 ± 0.08 ^BC^	0.95 ± 0.08 ^B^	0.25 ± 0.18 ^AB^	−0.05 ± 0.04 ^B^	−0.31 ± 0.14 ^B^
SR	0.88 ± 0.08 ^D^	1.21 ± 0.11 ^C^	1.23 ± 0.10 ^B^	0.33 ± 0.04 ^AB^	0.35 ± 0.17 ^A^	0.02 ± 0.20 ^A^
FT	2.30 ± 0.15 ^A^	1.74 ± 0.10 ^A^	1.75 ± 0.18 ^A^	−0.56 ± 0.05 ^D^	−0.55 ± 0.05 ^D^	0.01 ± 0.10 ^A^
DT	1.56 ± 0.16 ^B^	1.13 ± 0.08 ^C^	1.16 ± 0.05 ^B^	−0.43 ± 0.24 ^CD^	−0.41 ± 0.21 ^CD^	0.03 ± 0.03 ^A^
***Enterococcus gallinarum* CRBP19**						
Cryoprotector	Log reduction (log CFU/g 0%–log CFU/g 0.4%)	Log reduction (log CFU/g 0%–log CFU/g 0.7%)	Log reduction (log CFU/g pH 0%–log CFU/g 1%)	Log reduction (log CFU/g 0.4%–log CFU/g 0.7%)	Log reduction (log CFU/g 0.4%–log CFU/g 1%)	Log reduction (log CFU/g pH 0.7%–log CFU/g 1%)
CT	1.81 ± 0.10 ^A^	-	1.94 ± 0.08 ^A^	-	0.12 ± 0.17 ^A^	-
MD	1.14 ± 0.04 ^B^	-	-	-	-	-
SKM	0.09 ± 0.00 ^C^	0.26 ± 0.10 ^C^	0.12 ± 0.04 ^E^	0.16 ± 0.11 ^A^	0.03 ± 0.03 ^AB^	−0.13 ± 0.09 ^A^
TL	1.75 ± 0.04 ^A^	1.03 ± 0.08 ^B^	0.93 ± 0.08 ^D^	−0.71 ± 0.04 ^C^	−0.81 ± 0.08 ^C^	−0.10 ± 0.10 ^A^
SR	1.35 ± 0.05 ^B^	1.15 ± 0.16 ^AB^	1.27 ± 0.09 ^BC^	−0.20 ± 0.12 ^AB^	−0.08 ± 0.04 ^A^	0.12 ± 0.08 ^A^
FT	1.95 ± 0.09 ^A^	1.36 ± 0.12 ^A^	1.57 ± 0.23 ^B^	−0.59 ± 0.17 ^BC^	−0.38 ± 0.30 ^BC^	0.21 ± 0.30 ^A^
DT	1.91 ± 0.26 ^A^	1.28 ± 0.09 ^AB^	1.25 ± 0.06 ^C^	−0.62 ± 0.28 ^BC^	−0.66 ± 0.21 ^C^	−0.04 ± 0.08 ^A^

^A–E^ Superscript capital letters in the same column denote statistically different (*p* < 0.05) treatments of freeze-drying based on Tukey’s test. The values represent the mean ± standard deviation obtained from independent samples (*n* = 3). CT: control (phosphate-buffered saline—PBS); MD: maltodextrin; SKM: skim milk; TL: trehalose; SR: sucrose; FT: fructose; DT: dextrose. Negative values are interpreted as indicating the absence of loss of cell viability, suggesting tolerance after exposure to the stressful condition.

**Table 4 microorganisms-13-02486-t004:** Cell surface hydrophobicity (%) profiles of *Enterococcus faecium* CRBP46 and *Enterococcus gallinarum* CRBP19 freeze-drying with different cryoprotectants agents.

Cryoprotector	Cell Surface Hydrophobicity (%)	
	*Enterococcus faecium*	*Enterococcus gallinarum*
CT	29.86 ± 9.07 ^b^	14.65 ± 1.52 ^bc^
MD	39.90 ± 7.77 ^b^	15.53 ± 2.69 ^bc^
SKM	87.08 ± 2.81 ^a^	45.27 ± 5.56 ^a^
TL	37.30 ± 1.95 ^b^	18.32 ± 0.84 ^ab^
SR	39.22 ± 3.18 ^b^	17.83 ± 3.26 ^bc^
FT	42.13 ± 1.52 ^b^	11.24 ± 1.33 ^c^
DT	38.56 ± 1.69 ^b^	17.63 ± 0.38 ^ac^

Different lowercase letters in the same column denotes significant differences (*p* < 0.05) between treatments, based on the Games–Howell test. The values represent the mean ± standard deviation obtained from independent samples (*n* = 3). CT: control (phosphate-buffered saline—PBS); MD: maltodextrin; SKM: skim milk; TL: trehalose; SR: sucrose; FT: fructose; DT: dextrose.

## Data Availability

The original contributions presented in this study are included in the Article/Appendix A. Further inquiries can be directed to the corresponding authors.

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
