# Peer review of "Skim Milk as a Multifunctional Cryoprotectant for Fish Probiotic Enterococcus spp.: Impact on Viability During Lyophilization and Long-Term Storage"

_microorganisms, 2025, doi:10.3390/microorganisms13112486_

Round 1

Reviewer 1 Report

Comments and Suggestions for Authors

The manuscript entitled "Skim Milk As a Multifunctional Cryoprotectant for Fish Probiotic Enterococcus spp.: Impact on Viability During Lyophilisation and Long-Term Storage" is an interesting study with appropriate methods that could be improved, as well as well-written results and discussion. However, the methodology does not include enough methods to fully support the conclusions, as there are many additional methods that could be applied to encapsulated Enterococcus spp. (for example, zero-stress). There are also some other suggestions.

Line 46-Introduction should be shorter. 

Line165-Please, add the presure for lyophilisation for main and final drying.

Line 167-Did you determine the cellular viability for each treatment immediately after the addition of the culture to the medium?

Line 283-Please explain how the cell count of Enterococcus gallinarum increased after freeze-drying in trehalose.

Line 334-Please explain how you obtained a negative reduction of both Enterococcus sp. freeze-dried in different cryoprotectants after treatment at pH 5 and pH 3.

Line 354-Similar question, explan negative reduction for E. galinarium for example SKM after treatment with bile salts?

Line 391-Did you confimed with some other method that E. faecium produce EPS? 

Author Response

Reviewer 1:

Comment 1: The manuscript entitled "Skim Milk As a Multifunctional Cryoprotectant for Fish Probiotic Enterococcus spp.: Impact on Viability During Lyophilisation and Long-Term Storage" is an interesting study with appropriate methods that could be improved, as well as well-written results and discussion. However, the methodology does not include enough methods to fully support the conclusions, as there are many additional methods that could be applied to encapsulated Enterococcus spp. (for example, zero-stress). There are also some other suggestions.

Response 1: We would like to express our deep thanks to reviewer, for the valuable comments and the affirmation to our study. We have revised the manuscript following the comments.

Comment 2: Line 46-Introduction should be shorter. 

Response 2: We have revised the text and removed some general background information from the introduction, aiming to retain only the information essential for understanding the study. However, other specific details were added at the request of another reviewer (such as the in-depth safety context for the strains). Therefore, we have tried to balance the length of the introduction to the best of our ability.

Comment 3: Line165-Please, add the pressure for lyophilisation for main and final drying.

Response 3: We have updated the text as suggested by the reviewer (Please see lines: 178-180).

Comment 4: Line 167-Did you determine the cellular viability for each treatment immediately after the addition of the culture to the medium?

Response 4: After homogenising the cryoprotectants (20% w/v) in each suspension cellular, they were kept at room temperature (25 °C) for 30 min. This step is essential to allow cell permeation and reduce osmotic shock before freezing (Hubalek, 2003). After this period, we serially diluted each treatment to determine the number of colony-forming units. These results are shown in Table 1. The text has been modified to make this information clearer (Please see the lines: 170-175).

  • HUBALEK, Zdenek. Protectants used in the cryopreservation of microorganisms. Cryobiology, v. 46, n. 3, p. 205-229, 2003.

Comment 5: Line 283-Please explain how the cell counts of Enterococcus gallinarum increased after freeze-drying in trehalose.

Response 5: The apparent increase in E. gallinarum cell count after lyophilisation with trehalose (from 9.51 ± 0.03 to 9.54 ± 0.02) does not represent bacterial growth. The standard deviations of the results before and after lyophilisation overlap, indicating that any differences between the means are within the expected range. Therefore, trehalose did not increase E. gallinarum cell counts after lyophilisation, but it did maintain viability after drying.

Comment 6: Line 334-Please explain how you obtained a negative reduction of both Enterococcus spp. freeze-dried in different cryoprotectants after treatment at pH 5 and pH 3.

Response 6: Presenting our results in log reduction (logarithmic reduction) allowed us to assess whether there is maintenance (logarithmic reduction close to zero or negative) or loss of viability after exposing our Enterococcus spp. strains to stressful conditions. These negative reductions occur when the viable cell count after exposure to a stressful condition (e.g., pH 3) is slightly higher than the count under a milder condition (pH 5). However, we emphasise that this slight variation falls within the expected range of experimental variability inherent in the serial dilution methodology and CFU counts. Therefore, we interpret any negative or near-zero result as indicating that our cryoprotectants, especially skim milk, provided high stability after exposure to stressful conditions. We would also like to emphasise that, although this lack of viability reduction is not the most common result when exposing bacterial cultures to stressful conditions (where a positive reduction is generally expected, as observed in most of our results), it reinforces the excellent efficacy of cryoprotectants in our study. We have added a legend to Tables 2 and 3 to clarify this information for readers (Please see lines: 366-367; 389-391).

Comment 7: Line 354-Similar question, explain negative reduction for E. galinarium for example SKM after treatment with bile salts?

Response 7: The explanation is the same as in Comment 6, but applied here to bile salt stress. The "negative reduction" result observed for E. gallinarum with SKM follows the same principle: the viable cell count after exposure to a stressful condition (e.g., 0.7% or 1% bile salts) is slightly higher than the count under a milder condition (0.4% bile salts). As explained in the previous response, this minimal variation does not represent growth, but rather a relatively normal experimental variability of the plating method. Therefore, we interpret this result as the most substantial evidence of protection, as it protected bacterial cells at all bile salt concentrations. This reinforces the excellent efficacy of SKM in protecting cells against the effects of bile salts.

Comment 8: Line 391-Did you confimed with some other method that E. faecium produce EPS? 

Response 8: Financial limitations prevented us from performing more in-depth analyses to explore the EPS production by our Enterococcus strains. However, we received technical and scientific support from experienced researchers at the Multi-user Centre for the Analysis of Biomedical Phenomena, who assisted us in processing and morphologically describing the coating matrices used in our study, and consequently, in identifying EPS-like structures secreted in the extracellular matrix.

Lactic acid bacteria are microorganisms recognised for the production of exopolysaccharides, especially when subjected to stressful conditions, such as lyophilisation (Nguyen et al., 2020). In the literature, other well-designed studies and published in high-impact journals also used SEM as an alternative and effective tool for the identification of EPS for the strains Lactobacillus plantarum 201 (Guedes et al., 2019), L. acidophilus 05, Lacticaseibacillus casei 01 (Mafaldo et al., 2022), and L. casei (Guergoletto et al., 2010) subjected to freeze-drying or vacuum drying, respectively. Therefore, based on our findings, we believe that the structures observed in our study are EPS secreted by Enterococcus spp. However, we have updated the text in subsection 3.6 and section 4 to highlight that we observed EPS-like structures in the extracellular matrix (Please see lines: 425-427; 750-752) We plan to quantify EPS production in future studies.

  • NGUYEN, Phu-Tho et al. Exopolysaccharide production by lactic acid bacteria: the manipulation of environmental stresses for industrial applications. AIMS microbiology, v. 6, n. 4, p. 451, 2020.
  • GUEDES, Jéssica da Silva et al. Protective effects of β-glucan extracted from spent brewer yeast during freeze-drying, storage and exposure to simulated gastrointestinal conditions of probiotic lactobacilli. Lwt, v. 116, p. 108496, 2019.
  • MAFALDO, Ísis Meireles et al. Survival during long-term storage, membrane integrity, and ultrastructural aspects of Lactobacillus acidophilus 05 and Lacticaseibacillus casei 01 freeze-dried with freshwater microalgae biomasses. Food Research International, v. 159, p. 111620, 2022.
  • GUERGOLETTO, Karla Bigetti et al. Survival of Lactobacillus casei (LC-1) adhered to prebiotic vegetal fibers. Innovative Food Science & Emerging Technologies, v. 11, n. 2, p. 415-421, 2010.

Reviewer 2 Report

Comments and Suggestions for Authors

The article is very interesting in terms of investigating cryoprotectants for the preservation of probiotic strains. However, there are several questions for the authors.

Please explain why glycerol was not used as a cryoprotectant. It is one of the most common cryoprotectants and could have been used as a standard.

In section 2.3, you write that the cryoprotectants were mixed with the cell suspension. Please specify the characteristics of this suspension, including the initial number of cells.

In Figure 1, there is a clear increase in the number of enterococci in some experimental variants even after freezing. This likely occurred due to an initial difference in cell counts in the samples. Was this variability taken into account when calculating survival rates and their statistical significance?

In the discussion section, the authors describe the protective effect of milk by stating that milk components form a protective layer around the cells. The same explanation is given for the increase in hydrophobicity. I fully agree with the authors on this point. However, it seems that the discussion section should elaborate on this topic in more detail based on literature data. Please provide a more detailed review of what is currently known on this subject from the literature, adding 1–2 additional paragraphs.

The authors also note that "through differential interference contrast (DIC) microscopy, the cells of both bacteria were clearly observed inside the matrices." Please provide these data. If their volume is excessively large or repetitive, it is acceptable to include them in an Appendix rather than in the main text.

Author Response

Reviewer 2:

Comment 1: The article is very interesting in terms of investigating cryoprotectants for the preservation of probiotic strains. However, there are several questions for the authors.

Response 1: We would like to thank the reviewer for their positive comments and valuable suggestions, which have helped us improve our manuscript. We have revised the manuscript following the comments.

Comment 2: Please explain why glycerol was not used as a cryoprotectant. It is one of the most common cryoprotectants and could have been used as a standard.

Response 2: Glycerol is an excellent cryoprotectant for protecting bacterial cells from the adverse effects of freezing. However, its use is limited or completely avoided for obtaining dried bacterial cultures. Glycerol is a viscous cryoprotectant with a low freezing point, meaning it does not sublimate easily under the vacuum used in freeze-drying (Cornejo et al., 2017). Therefore, this incomplete drying results in a final product with residual moisture, which can compromise the stability and viability of the bacteria, especially at room temperature. Furthermore, glycerol also results in a viscous and sticky product.

Recently, Fredua-Agyeman et al. (2024) stated that glycerol is an excellent cryoprotectant for freezing bacteria; however, it is not recommended for use in freeze-drying. Similarly, POP et al. (2015) highlighted that the high viscosity of glycerol reduced freeze-drying efficiency.

  • FREDUA-AGYEMAN, Mansa. Surviving process and transit: Controlled freeze drying, storage, and enteric-coated capsules for targeted delivery of probiotic Lactobacillus acidophilus. Heliyon, v. 10, n. 7, 2024.
  • POP, Oana Lelia et al. Effect of glycerol, as cryoprotectant in the encapsulation and freeze drying of microspheres containing probiotic cells. Univ. Agric. Sci. Vet. Med. Cluj-Napoca. Food Sci. Technol, v. 72, p. 27-32, 2015.
  • CORNEJO, A. et al. Oxygenated fuel additives from glycerol valorization. Main production pathways and effects on fuel properties and engine performance: A critical review. Renewable and Sustainable Energy Reviews, v. 79, p. 1400-1413, 2017.

Comment 3: In section 2.3, you write that the cryoprotectants were mixed with the cell suspension. Please specify the characteristics of this suspension, including the initial number of cells.

Response 3: The standard cell suspension (SCS) consists of freshly harvested Enterococcus spp. bacterial cells by centrifugation and resuspended in sterile PBS (pH 7.4). This procedure was performed individually for each probiotic strain. The text has been modified to clarify this information (Please see lines: 166-167; 170-175). Additionally, the results of the initial viability counts for each cell suspension with cryoprotectant are presented in Table 1. Furthermore, we also emphasize that the viability presented in the control group corresponds to basal SCS, as it does not contain cryoprotectant.

Comment 4: In Figure 1, there is a clear increase in the number of enterococci in some experimental variants even after freezing. This likely occurred due to an initial difference in cell counts in the samples. Was this variability taken into account when calculating survival rates and their statistical significance?

Response 4: The bacterial cell counts results at time 0 refer to viability after freeze-drying. This fact explains the variation observed in the initial bacterial cell number for each treatment, since each cryoprotectant provided a different response for each Enterococcus strain. The logarithmic reduction at three temperatures for 120 days of storage was carefully analysed based on the logarithmic reduction at each storage time and for each cryoprotectant. This methodology involves individually evaluating the reduction in the number of viable Enterococcus cells for each treatment throughout storage, taking into account normal and expected variations in the number of bacterial cells at time 0 (after freeze-drying). Thus, by knowing the extent of reduction or maintenance achieved in each treatment, we can make more assertive statistical comparisons and determine which cryoprotectant provided the best protection for our lactic acid bacteria. This information is provided in section 2.5 of the manuscript (Please see the lines: 216-220).

Comment 5: In the discussion section, the authors describe the protective effect of milk by stating that milk components form a protective layer around the cells. The same explanation is given for the increase in hydrophobicity. I fully agree with the authors on this point. However, it seems that the discussion section should elaborate on this topic in more detail based on literature data. Please provide a more detailed review of what is currently known on this subject from the literature, adding 1–2 additional paragraphs.

Response 5: We have restructured the paragraph to explore the multifunctional properties of skim milk in more depth and with updated literature. The text has been updated according to the reviewer's suggestions (Please see lines: 675-690)

Comment 6: The authors also note that "through differential interference contrast (DIC) microscopy, the cells of both bacteria were clearly observed inside the matrices." Please provide these data. If their volume is excessively large or repetitive, it is acceptable to include them in an Appendix rather than in the main text.

Response 6: The data have been made available as supplementary material (Please see Figures S1 and S2). However, we highlight that the images have low resolution, as DIC analysis was used as a complementary tool to confirm the presence of bacterial cells within the matrices.

Reviewer 3 Report

Comments and Suggestions for Authors

Summary and Overall Assessment

The study comprehensively evaluates the impact of these agents on post-lyophilization viability, long-term storage stability at various temperatures, tolerance to simulated gastrointestinal fluids (SGF), and cell surface hydrophobicity. The central and most compelling finding is that skim milk (SKM) provides superior, multi-faceted protection, maintaining high viability under all tested stress conditions, most notably during storage at room temperature, where other agents failed significantly.

The study addresses a topic of significant practical importance for the advancement of sustainable aquaculture—the development of stable, host-specific probiotic formulations. The dataset presented is comprehensive, and the experimental design is logical in its approach to assessing multiple facets of probiotic functionality beyond simple survival. The work has the potential to contribute valuable knowledge to the field.

However, the manuscript in its current form suffers from several major deficiencies that are sufficiently serious to preclude its publication without substantial revision. These include a critical lack of justification and safety discussion regarding the use of Enterococcus species as probiotics, a superficial mechanistic explanation for the observed cryoprotective effects, and insufficient methodological detail that fundamentally compromises experimental reproducibility. These issues collectively weaken the scientific foundation and the broader applicability of the findings.

Detailed evaluation

  1. Critical Justification and Safety Considerations for the Use of Enterococcus Strains

The manuscript proposes the use of E. faecium and E. gallinarum as probiotics for aquaculture, a choice that is presented without adequate justification or discussion of the significant safety concerns associated with this genus. This omission is a fundamental flaw that must be rectified. The genus Enterococcus is not considered "Generally Recognized as Safe" (GRAS) by regulatory bodies and is a well-documented source of opportunistic nosocomial infections in humans.

Furthermore, enterococci are known for their intrinsic and acquired antibiotic resistance, acting as reservoirs for antibiotic resistance genes (ARGs) that can be transferred to other bacteria, including pathogens. The failure to include primary safety data within this manuscript.

More alarmingly, the application of these strains in aquaculture introduces a significant environmental and public health dimension that the authors have completely ignored. Probiotics administered to fish in aquaculture systems, which are often open or semi-open, can be released into the aquatic environment. This creates a potential reservoir for the dissemination of ARGs and virulence factors, which could ultimately impact the human food chain and contribute to the broader problem of antimicrobial resistance. The potential for gene transfer from a probiotic Enterococcus strain to other microbes in the gut or the environment represents a serious risk that cannot be overlooked. The authors must demonstrate that they have rigorously assessed and mitigated these risks.  

Required Revisions:

Introduction: A new paragraph must be added to the introduction that explicitly acknowledges the controversial nature of Enterococcus as a probiotic source. This section should transparently discuss the potential risks (e.g., pathogenicity, virulence factors, ARG transfer) and set the stage for how the authors have addressed these concerns for their specific strains, CRBP46 and CRBP19.

Materials and Methods / Discussion:

The authors must incorporate a detailed summary of the safety characterization of their strains directly into this manuscript. This is non-negotiable for a study proposing a new probiotic product. This summary should include:

Hemolysis Assay Results: Data confirming the absence of β-hemolytic activity, a primary screen for cytotoxicity.

Antibiotic Susceptibility Profile: A comprehensive antibiogram, tested against a panel of clinically relevant antibiotics as recommended by EFSA guidelines. This must include, but not be limited to, ampicillin and vancomycin, as resistance to these agents is a major red flag.

Genetic Screening: Results from PCR or whole-genome screening for the presence of known virulence factor genes (e.g., gelE, esp, hyl, ace) and key transferable ARGs (e.g., van genes for vancomycin resistance).

This essential safety data must be presented clearly, either in the main text or in a supplementary table, and must be thoroughly discussed to build a convincing, evidence-based case for the safety of these specific isolates for their intended application.

  1. Insufficient Mechanistic Elucidation for Cryoprotectant Efficacy

The Discussion section attributes the superiority of skim milk (SKM) primarily to its ability to form a dense matrix with low porosity, an observation derived from scanning electron microscopy (SEM). While this physical characteristic is relevant, it represents a superficial explanation that fails to explore the complex and multi-faceted biochemical mechanisms that underpin effective cryoprotection.

The manuscript should discuss SKM not as a monolithic substance but as a multi-component, synergistic protective system. It’s remarkable efficacy stems from the combined and interactive effects of its constituents.

A thorough discussion should have explored the distinct roles of:  

Milk Proteins (Caseins and Whey): These proteins form a physical coating around the bacterial cells, which stabilizes the cell membrane against mechanical stress from ice crystals. They can also form hydrophobic bonds with the bacterial surface, which likely contributes to the observed increase in cell surface hydrophobicity, a property linked to better mucosal adhesion.

Lactose: This disaccharide is a key player that functions via the "water replacement hypothesis." During dehydration, lactose molecules form hydrogen bonds with the polar headgroups of membrane phospholipids, effectively replacing bound water and preventing damaging phase transitions that lead to membrane leakage and loss of function.

Minerals (e.g., Calcium): Divalent cations like Ca2+ present in milk can stabilize subcellular structures and interact with milk proteins to form protective agglomerates that encapsulate the cells.

Buffering Capacity: The high concentration of proteins in SKM provides significant buffering capacity. This property almost certainly explains the superior performance of SKM-protected cells in the acidic simulated gastric fluid test, a point not made by the authors.

Antioxidant Properties: Cryopreservation processes are known to induce oxidative stress, leading to DNA and membrane damage. SKM has been shown to reduce such oxidative damage, a critical protective mechanism that is entirely absent from the current discussion.

The most significant weakness of the discussion is its failure to conduct a comparative mechanistic analysis. Trehalose, which performed well immediately post-lyophilization but failed dramatically during room-temperature storage, provides a perfect scientific contrast to SKM. A good quality discussion would have contrasted SKM's multi-modal protection with trehalose's primary mechanisms, which are vitrification (the formation of a stable, non-crystalline glassy matrix that immobilizes cells and prevents ice crystal growth) and water replacement. A deeper analysis would then hypothesize why trehalose failed at 25℃. The glassy state of trehalose is highly sensitive to temperature and ambient humidity. Above its glass transition temperature (Tg), which can be lowered by moisture absorption, the matrix can collapse from a rigid glass to a rubbery state, losing its protective properties and exposing cells to moisture, oxidation, and chemical degradation. In contrast, the robust protein-polysaccharide matrix of SKM likely provides superior physical protection against ambient humidity and temperature fluctuations, explaining its exceptional long-term stability at room temperature.

Required Revisions:

The Discussion section must provide a deep, comparative mechanistic analysis. It should include:

A detailed breakdown of the multiple, synergistic protective mechanisms afforded by SKM's various components (proteins, lactose, minerals, etc.).

A direct comparison of these mechanisms to those of the other tested cryoprotectants, particularly trehalose, explaining their relative successes and failures under different conditions.

A well-reasoned hypothesis, grounded in the principles of cryobiology and materials science, to explain why SKM excelled in long-term room-temperature storage and gastrointestinal tolerance while other agents, notably trehalose, did not.

  1. Lack of Methodological Detail for Experimental Reproducibility

The lyophilization procedure is described with only three parameters: a pre-freezing step at −80℃ for 2 hours, a total cycle duration of 40 hours, and a condenser temperature of −50℃. This is critically insufficient.

Key parameters that dictate the physics of ice crystal formation and sublimation, and thus the ultimate survival of the microorganisms, are completely absent. These include the cooling rate during freezing, the shelf temperatures during primary and secondary drying phases, and the precise vacuum levels maintained during each phase.

This lack of detail extends to the storage stability experiment, which is central to the paper's main claims. The manuscript does not specify the type of cryotubes used, how they were sealed (e.g., screw cap tightness, use of parafilm), or the ambient relative humidity during the storage period. Moisture is the primary cause of degradation for lyophilized products. Without controlling and reporting these variables, the significant viability loss observed at 25℃ for most treatments could be an artifact of poor sample storage (e.g., moisture ingress due to inadequate sealing) rather than a true reflection of the cryoprotectant's intrinsic efficacy.

The superior stability of the SKM formulation at room temperature. The observed differences could be due to differences in the hygroscopicity of the various matrices or simply due to inconsistent sealing across samples. The current methodology makes it impossible to distinguish between these possibilities.  

Required Revisions:

The authors must add a table to the Materials and Methods section detailing the complete, multi-step lyophilization cycle. This table should provide a precise, replicable protocol.

Phase

Parameter

Value

Freezing

Shelf Temperature (℃)

Cooling Rate (℃/min)

Hold Time (min)

Primary Drying

Chamber Pressure (mTorr or Pa)

Shelf Temperature (℃)

Ramp Rate (℃/min)

Hold Time (hours)

Secondary Drying

Chamber Pressure (mTorr or Pa)

Shelf Temperature (℃)

Ramp Rate (℃/min)

Hold Time (hours)

Specific Section-by-Section Comments

Introduction

Insufficient Ecological Context: The introduction fails to adequately describe the native gut microbiome of Arapaima gigas. Existing research indicates that the gut of this carnivorous fish is often dominated by genera like Cetobacterium (phylum Fusobacteria), with Firmicutes (the phylum containing Enterococcus) also being a significant component. The authors should include this information to properly contextualize their work. Providing this context is vital for justifying the "autochthonous" approach and for discussing the potential ecological impact of supplementing these specific strains.  

Rationale for Strain Selection: Beyond the major safety concerns detailed above, the rationale for selecting these specific Enterococcus strains is not well-supported by a broad literature review within the introduction. The section should be strengthened by connecting the known probiotic traits of certain Enterococcus strains (e.g., high resilience to environmental stress, production of bacteriocins) to the specific health challenges in arapaima aquaculture (e.g., susceptibility to common pathogens like Aeromonas hydrophila, which has been documented in this fish).  

Materials and Methods

Statistical Analysis (Section 2.9): The stated use of "Welch's ANOVA" for non-parametric data is incorrect. Welch's ANOVA is an adaptation of the standard one-way ANOVA used for parametric data when the assumption of equal variances is violated. The appropriate non-parametric alternative for comparing more than two independent groups is the Kruskal-Wallis test, which should be followed by a suitable post-hoc test (e.g., Dunn's test) for pairwise comparisons. This methodological error must be corrected.  

Method Validation: The manuscript states a detection limit of 2 log CFU/g but provides no validation data to support this claim. For the plate counting method, the authors should briefly describe how this limit of detection (LOD) and the limit of quantification (LOQ) were established to ensure methodological rigor and adherence to quality standards.  

Results

Interpretation of Results: The text often presents data without sufficient immediate interpretation. For instance, in section 3.5 (Bacterial Adhesion), the authors state that SKM rendered the strains "moderately hydrophobic or hydrophobic". They should also state the practical implication directly within the results section, for example: "This marked increase in hydrophobicity suggests that lyophilization with SKM may enhance the strains' potential to adhere to the gut mucosa, a key prerequisite for colonization." This practice helps to bridge the raw data to its biological meaning for the reader.  

Discussion

Study Limitations: The discussion of the study's limitations needs to be more explicit and critical to provide a balanced perspective.

Storage Duration: A 120-day storage period is insufficient to make broad claims about commercial shelf-life, which often requires demonstrated stability for 12-24 months. This should be clearly stated as a limitation and a direction for future work.

In Vitro vs. In Vivo: The study is entirely in vitro. The authors must explicitly state that while their results on gastrointestinal tolerance and hydrophobicity are promising indicators, they cannot be directly extrapolated to predict colonization, persistence, and efficacy in vivo within the complex and dynamic gut environment of a live arapaima. This is a critical caveat that tempers the interpretation of the findings.

Overstated Conclusions: The language in the Conclusions section is, at times, too strong and speculative. Phrases such as "ideal cryoprotectant" should be tempered to "the most effective cryoprotectant under the tested conditions." Claims of "economic, marketing, and even social benefits" are entirely unsubstantiated by the data presented in this study and should be removed. The conclusion must be a direct, measured, and evidence-based summary of the study's empirical findings, avoiding hyperbole.  

Author Response

Reviewer 3:

Comment 1: The study comprehensively evaluates the impact of these agents on post-lyophilization viability, long-term storage stability at various temperatures, tolerance to simulated gastrointestinal fluids (SGF), and cell surface hydrophobicity. The central and most compelling finding is that skim milk (SKM) provides superior, multi-faceted protection, maintaining high viability under all tested stress conditions, most notably during storage at room temperature, where other agents failed significantly.

The study addresses a topic of significant practical importance for the advancement of sustainable aquaculture—the development of stable, host-specific probiotic formulations. The dataset presented is comprehensive, and the experimental design is logical in its approach to assessing multiple facets of probiotic functionality beyond simple survival. The work has the potential to contribute valuable knowledge to the field.

Response 1: We thank the reviewer for their valuable comments and for recognising the relevance of our study. Indeed, the central objective of this study was to address a critical point in probiotic production: maintaining cell viability, even after exposure to various stressful conditions.

Comment 2: However, the manuscript in its current form suffers from several major deficiencies that are sufficiently serious to preclude its publication without substantial revision. These include a critical lack of justification and safety discussion regarding the use of Enterococcus species as probiotics, a superficial mechanistic explanation for the observed cryoprotective effects, and insufficient methodological detail that fundamentally compromises experimental reproducibility. These issues collectively weaken the scientific foundation and the broader applicability of the findings.

Response 2: We carefully considered each point raised by the reviewer and conducted an extensive review of our manuscript.

Detailed evaluation

  1. Critical Justification and Safety Considerations for the Use of Enterococcus Strains

Comment 3: The manuscript proposes the use of E. faecium and E. gallinarum as probiotics for aquaculture, a choice that is presented without adequate justification or discussion of the significant safety concerns associated with this genus. This omission is a fundamental flaw that must be rectified. The genus Enterococcus is not considered "Generally Recognized as Safe" (GRAS) by regulatory bodies and is a well-documented source of opportunistic nosocomial infections in humans.

Furthermore, enterococci are known for their intrinsic and acquired antibiotic resistance, acting as reservoirs for antibiotic resistance genes (ARGs) that can be transferred to other bacteria, including pathogens. The failure to include primary safety data within this manuscript.

More alarmingly, the application of these strains in aquaculture introduces a significant environmental and public health dimension that the authors have completely ignored. Probiotics administered to fish in aquaculture systems, which are often open or semi-open, can be released into the aquatic environment. This creates a potential reservoir for the dissemination of ARGs and virulence factors, which could ultimately impact the human food chain and contribute to the broader problem of antimicrobial resistance. The potential for gene transfer from a probiotic Enterococcus strain to other microbes in the gut or the environment represents a serious risk that cannot be overlooked. The authors must demonstrate that they have rigorously assessed and mitigated these risks.  

Response 3: We agree with the reviewer and have conducted an extensive review of the manuscript to clarify the issues raised by the reviewer. 

Required Revisions:

Comment 4: Introduction: A new paragraph must be added to the introduction that explicitly acknowledges the controversial nature of Enterococcus as a probiotic source. This section should transparently discuss the potential risks (e.g., pathogenicity, virulence factors, ARG transfer) and set the stage for how the authors have addressed these concerns for their specific strains, CRBP46 and CRBP19.

Response 4: We understand and share the concern (both theoretical and practical) regarding the controversial nature of the Enterococcus genus. We have updated the text (Please see lines 101-104) and now explicitly acknowledge the potential risks (e.g., pathogenicity, ARG transfer), based on the reviewer's recommendations.

We also clarify that our specific strains (CRBP46 and CRBP19) were chosen after an in vitro screening (e.g., hemolytic activity, antagonistic activity against pathogenic bacteria, and antimicrobial susceptibility profile). Furthermore, we also performed a safety assay by intraperitoneally inoculating high concentrations (~10^8 CFU/mL) of these bacteria into healthy arapaima. These analyses were carried out in a parallel study and have been added to the text as [Reference 30]. We would like to emphasize that this article does not aim to recommend the immediate use of these bacteria for aquaculture, but rather to study one of the critical steps in probiotic development (i.e., viability during the production and storage process). Therefore, we believe this addition provides the crucial safety context the reviewer requested, while allowing the current manuscript to remain focused on its specific biotechnological objective.

Comment 5: Materials and Methods / Discussion:

The authors must incorporate a detailed summary of the safety characterization of their strains directly into this manuscript. This is non-negotiable for a study proposing a new probiotic product. This summary should include:

Hemolysis Assay Results: Data confirming the absence of β-hemolytic activity, a primary screen for cytotoxicity.

Antibiotic Susceptibility Profile: A comprehensive antibiogram, tested against a panel of clinically relevant antibiotics as recommended by EFSA guidelines. This must include, but not be limited to, ampicillin and vancomycin, as resistance to these agents is a major red flag.

Genetic Screening: Results from PCR or whole-genome screening for the presence of known virulence factor genes (e.g., gelE, esp, hyl, ace) and key transferable ARGs (e.g., van genes for vancomycin resistance).

This essential safety data must be presented clearly, either in the main text or in a supplementary table, and must be thoroughly discussed to build a convincing, evidence-based case for the safety of these specific isolates for their intended application.

Response 5: We thank the reviewer for the comment. The study that aimed to screen and characterize potential probiotics for arapaima is still in the process of submission/publication (Reference 30). In this parallel study, we address in vitro and in vivo tests as tools for selecting potential probiotic strains for different South American fish species (namely, Colossoma macropomum, Brycon amazonicus, and Arapaima gigas), as well as their limitations.

Thus, as it is still in the publication process, we decided not to include an in-depth discussion of these data, as this was not the objective of the present paper. We agree with the reviewer, and we have added pertinent information regarding the safety of these strains for arapaima (Please see lines: 104-108; 152-156).

Again, we understand the reviewer's concern and completely agree that the development of commercial probiotics for any animal production must be done with caution. Therefore, we believe that the advancement of probiotic development for fish is linked to the generation of scientific knowledge in different areas. Here, we focus on the biotechnological process of developing a stable probiotic formulation, which includes selecting the most effective cryoprotectant for the different conditions tested.

Comment 6: 2. Insufficient Mechanistic Elucidation for Cryoprotectant Efficacy

The Discussion section attributes the superiority of skim milk (SKM) primarily to its ability to form a dense matrix with low porosity, an observation derived from scanning electron microscopy (SEM). While this physical characteristic is relevant, it represents a superficial explanation that fails to explore the complex and multi-faceted biochemical mechanisms that underpin effective cryoprotection.

The manuscript should discuss SKM not as a monolithic substance but as a multi-component, synergistic protective system. It’s remarkable efficacy stems from the combined and interactive effects of its constituents.

Response 6: We thank the reviewer for their valuable suggestions. Below, we have made significant changes to the manuscript based on the reviewer's recommendations.

Comment 7: A thorough discussion should have explored the distinct roles of:  

Milk Proteins (Caseins and Whey): These proteins form a physical coating around the bacterial cells, which stabilizes the cell membrane against mechanical stress from ice crystals. They can also form hydrophobic bonds with the bacterial surface, which likely contributes to the observed increase in cell surface hydrophobicity, a property linked to better mucosal adhesion.

Response 7: We added the information in the text following the reviewer's recommendations (Please see lines: 711-718).

Comment 8: Lactose: This disaccharide is a key player that functions via the "water replacement hypothesis." During dehydration, lactose molecules form hydrogen bonds with the polar headgroups of membrane phospholipids, effectively replacing bound water and preventing damaging phase transitions that lead to membrane leakage and loss of function.

Response 8: We have updated the text as suggested by the reviewer (Please see lines: 683-685).

Comment 9: Minerals (e.g., Calcium): Divalent cations like Ca2+ present in milk can stabilize subcellular structures and interact with milk proteins to form protective agglomerates that encapsulate the cells.

Response 9: We corrected as suggested by the reviewer (Please see lines: 686-688).

Comment 10: Buffering Capacity: The high concentration of proteins in SKM provides significant buffering capacity. This property almost certainly explains the superior performance of SKM-protected cells in the acidic simulated gastric fluid test, a point not made by the authors.

Response 10: We have updated the text as suggested by the reviewer (Please see lines: 668-670).

Comment 11: Antioxidant Properties: Cryopreservation processes are known to induce oxidative stress, leading to DNA and membrane damage. SKM has been shown to reduce such oxidative damage, a critical protective mechanism that is entirely absent from the current discussion.

Response 11: We added the information in the text following the reviewer's recommendations (Please see lines: 676-679).

Comment 12: The most significant weakness of the discussion is its failure to conduct a comparative mechanistic analysis. Trehalose, which performed well immediately post-lyophilization but failed dramatically during room-temperature storage, provides a perfect scientific contrast to SKM. A good quality discussion would have contrasted SKM's multi-modal protection with trehalose's primary mechanisms, which are vitrification (the formation of a stable, non-crystalline glassy matrix that immobilizes cells and prevents ice crystal growth) and water replacement. A deeper analysis would then hypothesize why trehalose failed at 25℃. The glassy state of trehalose is highly sensitive to temperature and ambient humidity. Above its glass transition temperature (Tg), which can be lowered by moisture absorption, the matrix can collapse from a rigid glass to a rubbery state, losing its protective properties and exposing cells to moisture, oxidation, and chemical degradation. In contrast, the robust protein-polysaccharide matrix of SKM likely provides superior physical protection against ambient humidity and temperature fluctuations, explaining its exceptional long-term stability at room temperature.

Response 12: We corrected as suggested by the reviewer (Please see lines: 611-632).

Comment 13: Required Revisions

The Discussion section must provide a deep, comparative mechanistic analysis. It should include:

A detailed breakdown of the multiple, synergistic protective mechanisms afforded by SKM's various components (proteins, lactose, minerals, etc.).

A direct comparison of these mechanisms to those of the other tested cryoprotectants, particularly trehalose, explaining their relative successes and failures under different conditions.

A well-reasoned hypothesis, grounded in the principles of cryobiology and materials science, to explain why SKM excelled in long-term room-temperature storage and gastrointestinal tolerance while other agents, notably trehalose, did not.

Response 13: We thank the reviewer for their insightful and constructive comments. The reviewer provided invaluable input into the discussion of the article. We agree with every point the reviewer raised and have carefully revised our manuscript, aiming to provide a thorough and insightful discussion for readers.

Comment 14: 3. Lack of Methodological Detail for Experimental Reproducibility

The lyophilization procedure is described with only three parameters: a pre-freezing step at −80℃ for 2 hours, a total cycle duration of 40 hours, and a condenser temperature of −50℃. This is critically insufficient. Key parameters that dictate the physics of ice crystal formation and sublimation, and thus the ultimate survival of the microorganisms, are completely absent. These include the cooling rate during freezing, the shelf temperatures during primary and secondary drying phases, and the precise vacuum levels maintained during each phase.

Response 14: We agree with the reviewer that detailing the freeze-drying parameters is important for the reproducibility of the method. Unfortunately, the equipment used in our study was a benchtop lyophilizer (LJJ, JJ Científica, São Paulo, Brazil), which does not have sensors or programmable controls for the cooling rate or shelf temperature during the drying phases.

To best meet the reviewer's request and to the best of our ability, we have added information on the vacuum pressure maintained throughout the process to the manuscript (please see lines: 178-180). Our equipment measures vacuum pressure in mbar, however, we made a conversion to mTorr and Pa, in order to meet the reviewer's suggestion. We hope that adding the vacuum pressure, along with the context of the equipment used, will help to detail and strengthen the description of our protocol. 

Comment 15: This lack of detail extends to the storage stability experiment, which is central to the paper's main claims. The manuscript does not specify the type of cryotubes used, how they were sealed (e.g., screw cap tightness, use of parafilm), or the ambient relative humidity during the storage period. Moisture is the primary cause of degradation for lyophilized products. Without controlling and reporting these variables, the significant viability loss observed at 25℃ for most treatments could be an artifact of poor sample storage (e.g., moisture ingress due to inadequate sealing) rather than a true reflection of the cryoprotectant's intrinsic efficacy.

The superior stability of the SKM formulation at room temperature. The observed differences could be due to differences in the hygroscopicity of the various matrices or simply due to inconsistent sealing across samples. The current methodology makes it impossible to distinguish between these possibilities.  

Response 15: We conducted an extensive review of the text to carefully address all of the reviewer's recommendations (please see lines: 180-183; 211-213).

Specific Section-by-Section Comments

Introduction

Comment 16: Insufficient Ecological Context: The introduction fails to adequately describe the native gut microbiome of Arapaima gigas. Existing research indicates that the gut of this carnivorous fish is often dominated by genera like Cetobacterium (phylum Fusobacteria), with Firmicutes (the phylum containing Enterococcus) also being a significant component. The authors should include this information to properly contextualize their work. Providing this context is vital for justifying the "autochthonous" approach and for discussing the potential ecological impact of supplementing these specific strains.  

Response 16: We agree with the reviewer and have added this information to the introduction (please see lines: 111-113).

Comment 17: Rationale for Strain Selection: Beyond the major safety concerns detailed above, the rationale for selecting these specific Enterococcus strains is not well-supported by a broad literature review within the introduction. The section should be strengthened by connecting the known probiotic traits of certain Enterococcus strains (e.g., high resilience to environmental stress, production of bacteriocins) to the specific health challenges in arapaima aquaculture (e.g., susceptibility to common pathogens like Aeromonas hydrophila, which has been documented in this fish).  

Response 17: We agree with the reviewer and have added this information to the introduction (please see lines: 113-119).

Materials and Methods

Comment 18: Statistical Analysis (Section 2.9): The stated use of "Welch's ANOVA" for non-parametric data is incorrect. Welch's ANOVA is an adaptation of the standard one-way ANOVA used for parametric data when the assumption of equal variances is violated. The appropriate non-parametric alternative for comparing more than two independent groups is the Kruskal-Wallis test, which should be followed by a suitable post-hoc test (e.g., Dunn's test) for pairwise comparisons. This methodological error must be corrected.  

Response 18: We agree with the reviewer. We used Welch's ANOVA because our cell surface hydrophobicity results presented normal but showed heterogeneous variances, which prevented us from analysing them using a standard one-way ANOVA. Unfortunately, we made a textual error when describing this information in section 2.9. The text has been appropriately updated to improve data reproducibility (Please see lines: 286-295).

Comment 19: Method Validation: The manuscript states a detection limit of 2 log CFU/g but provides no validation data to support this claim. For the plate counting method, the authors should briefly describe how this limit of detection (LOD) and the limit of quantification (LOQ) were established to ensure methodological rigor and adherence to quality standards.  

Response 19: We agree with the reviewer and have added an explanation of the detection and quantification limits established in our study to subsection 2.3 (Please see lines: 186-193) to ensure methodological rigor and adherence to quality standards.

Results

Comment 20: Interpretation of Results: The text often presents data without sufficient immediate interpretation. For instance, in section 3.5 (Bacterial Adhesion), the authors state that SKM rendered the strains "moderately hydrophobic or hydrophobic". They should also state the practical implication directly within the results section, for example: "This marked increase in hydrophobicity suggests that lyophilization with SKM may enhance the strains' potential to adhere to the gut mucosa, a key prerequisite for colonization." This practice helps to bridge the raw data to its biological meaning for the reader.  

Response 20: This information has now been included in subsection 3.5 (Please see lines: 400-402).

Discussion

Comment 21: Study Limitations: The discussion of the study's limitations needs to be more explicit and critical to provide a balanced perspective.

Response 21: The limitations suggested by the reviewer were carefully considered and incorporated into the manuscript, aiming to provide a more balanced perspective on our main findings. 

Comment 22: Storage Duration: A 120-day storage period is insufficient to make broad claims about commercial shelf-life, which often requires demonstrated stability for 12-24 months. This should be clearly stated as a limitation and a direction for future work.

Response 22: We agree with the reviewer's observation and, following their recommendation, all statements about commercial validation were carefully removed from the text, and the study duration was included as a limitation in section 4 (Please see lines: 633-638). Our choice to evaluate viability for 120 days at three storage temperatures (-25°C, 4°C, and 25°C) was based on this being a period considered satisfactory for evaluations under laboratory conditions, and also on similar studies published in journals (Li et al., 2024; Tian et al., 2024; Benkirane et al., 2024; Araújo et al., 2025).

We would also like to emphasize that although we focused the comparison on 120 days of storage, our Enterococcus spp. strains remained viable for 365 days (12 months) under freezing conditions (-25 °C). This information was already presented in the text (Please see lines: 660-662), which reinforces the long-term stability for this storage condition.

  • LI, Xiao-min et al. An effective strategy for improving the freeze-drying survival rate of Lactobacillus curvatus and its potential protective mechanism. Food Bioscience, v. 58, p. 103794, 2024.
  • TIAN, Yue et al. Effect of freeze-dried protectants on the survival rate and fermentation performance of fermented milk's directed vat set starters. Cryobiology, v. 114, p. 104811, 2024.
  • BENKIRANE, Ghita et al. Effect of microencapsulation on the bio-preservative and probiotic properties of Enterococcus duransFood Bioscience, v. 60, p. 104312, 2024.
  • ARAÚJO, Alessandra Silva et al. Influence of the addition of gum arabic and xanthan gum in the preparation of sodium alginate microcapsules coated with chitosan hydrochloride on the survival of Lacticaseibacillus rhamnosusInternational Journal of Biological Macromolecules, v. 294, p. 139388, 2025.

Comment 23: In Vitro vs. In Vivo: The study is entirely in vitro. The authors must explicitly state that while their results on gastrointestinal tolerance and hydrophobicity are promising indicators, they cannot be directly extrapolated to predict colonization, persistence, and efficacy in vivo within the complex and dynamic gut environment of a live arapaima. This is a critical caveat that tempers the interpretation of the findings.

Response 23: We agree with the reviewer and have added a new paragraph (please see lines: 719-728) in section 4 of the manuscript.

Comment 24: Overstated Conclusions: The language in the Conclusions section is, at times, too strong and speculative. Phrases such as "ideal cryoprotectant" should be tempered to "the most effective cryoprotectant under the tested conditions." Claims of "economic, marketing, and even social benefits" are entirely unsubstantiated by the data presented in this study and should be removed. The conclusion must be a direct, measured, and evidence-based summary of the study's empirical findings, avoiding hyperbole.  

Response 24: The conclusion has been appropriately adjusted following the reviewer's recommendations (Please see lines: 36-40; 769-786).

Round 2

Reviewer 1 Report

Comments and Suggestions for Authors

The revisied version of manuscript entiteled Skim Milk As a Multifunctional Cryoprotectant for Fish Probiotic Enterococcus spp.: Impact on Viability During Lyophilization and Long-Term Storage,  is suitable for publication in Microorganisms. The authors made the necessary changes and give appropiate answers for comments.